# SERS discrimination of single DNA bases in single oligonucleotides by electro-plasmonic trapping

Jian-An Huang [1], Mansoureh Z. Mousavi[1], Yingqi Zhao[1], Aliaksandr Hubarevich[1], Fatima Omeis[1], Giorgia Giovannini[1], Moritz Schütte[2], Denis Garoli[1,3]* & Francesco De Angelis [1]*

Surface-enhanced Raman spectroscopy (SERS) sensing of DNA bases by plasmonic nanopores could pave a way to novel methods for DNA analyses and new generation single-molecule sequencing platforms. The SERS discrimination of single DNA bases depends critically on the time that a DNA strand resides within the plasmonic hot spot. In fact, DNA molecules flow through the nanopores so rapidly that the SERS signals collected are not sufficient for single-molecule analysis. Here, we report an approach to control the residence time of molecules in the hot spot by an electro-plasmonic trapping effect. By directly adsorbing molecules onto a gold nanoparticle and then trapping the single nanoparticle in a plasmonic nanohole up to several minutes, we demonstrate single-molecule SERS detection of all four DNA bases as well as discrimination of single nucleobases in a single oligonucleotide. Our method can be extended easily to label-free sensing of single-molecule amino acids and proteins.

[1] Istituto Italiano di Tecnologia, Via Morego 30, 16163 Genova, Italy. [2] Alacris Theranostics GmbH, Max-Planck-Straβe 3, D-12489 Berlin, Germany. [3] AB ANALITICA s.r.l., Via Svizzera 16, 35127 Padova, Italy. *email: Denis.Garoli@iit.it; Francesco.Deangelis@iit.it

Surface-enhanced Raman spectroscopy (SERS) allows label-free detection of single-analyte molecules by their narrow-fingerprint Raman peaks[1], which is promising for biomedical analysis such as single-molecule sequencing[2,3]. SERS methods usually employ plasmonic metal nanostructures with a sub-10-nm gap feature that, upon laser excitation, exhibit a strong localized electromagnetic field on their surface, termed hot spots. When molecules enter the hot spots, their Raman signals are excited and enhanced by the electromagnetic field so much that single-molecule sensitivity can be achieved[4].

SERS sensing based on a flow-through scheme is desirable for many practical applications including lab-on-a-chip diagnostics[5]. However, some challenges have to be addressed for this technology to evolve. Among them, it would be crucial to have control on the time that the analyte molecule resides in the plasmonic hot spot[2]. For example, DNA strands flow through solid-state nanopores so fast (a few microseconds) that SERS signals collected are not sufficient for single-molecule analysis[3]. Another recent work reported that SERS detection of nucleobases by plasmonic nanoslits could not achieve single-molecule resolution until a collection time of 100 ms was used[2].

In other words, the fast transport of the analyte molecules represents a current obstacle hindering the development of SERS-based flow-through sensors. Recently, different approaches were developed to integrate electrical, optical, and thermal forces to control plasmonic nano-object motion at the nanoscale[6–8] and to slow down molecule transport[9–11]. However, reports that combine these effects to demonstrate single-molecule SERS in the flow-through scheme are still missing. By overcoming this challenge, it may create a revolution of diagnostic devices in, for example, DNA sequencing that reached market level years ago by utilizing solid-state nanopores[12]. In comparison with those pioneering nanopore technologies, Raman-based sensors offer even more advantages, thanks to much higher discrimination power provided by Raman spectroscopy. For instance, each DNA base can be distinguished in a Raman spectrum of a DNA strand by its own narrow-fingerprint Raman peaks[13,14]. In principle, the capability of multiplex analysis by Raman spectroscopy may pave the way to single-molecule protein sequencing that is challenging, but highly desirable in both basic research and diagnostic applications[15].

In this work, we introduce an electro-plasmonic approach to control the residence time of biomolecules in a hot spot by trapping a gold nanourchin (AuNU) in a plasmonic nanohole. The physical mechanism relies on the balance between the electroosmotic, electrophoretic, and optical forces. To show the discrimination power of this approach and its potential applications to DNA and protein analysis, we chose nucleotides as molecules of interest. The AuNUs have many sharp tips that could confine electromagnetic fields for SERS detections[16–18]. Prior to trapping, the AuNUs are mixed in solution with the nucleotides to allow them to directly adsorb on the surface of the AuNUs. Once trapped, the sharp tips of the AuNU will couple with the sidewall of the nanohole creating plasmonic hot spots, which exhibit a greatly enhanced electromagnetic field for SERS detection of the nucleotides already adsorbed on the tips. As we show thereafter in detail, by applying electric potentials, we can keep the AuNUs trapped for minutes such that DNA bases can stay in the hot spots long enough to achieve single-molecule analysis.

## Results

### Plasmonic field induced by coupling of a gold nanohole with a AuNU.
A polydimethylsiloxane (PDMS) chamber is used to house a gold nanohole array on a silicon nitride (SiN) membrane to form a trans compartment and a cis compartment (Fig. 1a),

which allows electro-plasmonic trapping and SERS detection of the AuNUs upon laser excitation (Fig. 1b). The gold nanohole array with a hole size of 200 nm (Fig. 1c) is fabricated by deposition of a 100-nm-thick gold film onto a supporting 100-nm-thick SiN membrane before focused ion beam milling of the nanohole array. A sequential 5-nm layer of alumina is deposited by atomic layer deposition to control the sample's surface charge (see the "Methods" section for details). This nanohole diameter is chosen as it exhibits plasmonic resonance at around the 785-nm laser wavelength (Supplementary Fig. 1). Once the 50-nm AuNU (Fig. 1d and Supplementary Fig. 2) with adsorbed analyte molecules on its surface flows near the nanohole sidewall, upon laser excitation, the charge oscillation of the AuNU tip couples to the charge of the nanohole sidewall. This induces an intense, confined electromagnetic field at the AuNU tip (Fig. 1e, f) for SERS sensing[19]. The closer the AuNU tip is to the sidewall, the narrower and stronger the electromagnetic field is found to be (Supplementary Fig. 3). As a result, single-molecule detection can be achieved by controlling the proximity between the AuNU and the nanohole's wall as illustrated in Fig. 1b.

**Electro-plasmonic trapping effect.** The confinement of the AuNU near the nanohole wall is due to electro-plasmonic trapping, which is a combination of electrokinetic and optical forces (Fig. 2a)[6,20]. Hence, the correct name for this effect should be electro-opto-kinetic to take into account the three components of the mechanism. For the sake of simplicity, we call it electro-plasmonic trapping. Without a bias applied across the nanohole, the AuNUs ($10^{10}$ particles per mL) in the cis chamber will diffuse to the trans chamber filled only with solvent, driven by entropic force due to the concentration gradient ($1:10^{10}$) and Brownian motion. When an AuNU diffuses through the nanohole upon laser illumination, the optical force due to the gradient of the plasmonic resonant electromagnetic field of the nanohole pulls the AuNU to the sidewall. Moreover, when the AuNU is pulled close to the sidewall, the coupling between the nanohole and the AuNU enhances the electromagnetic field as well as the gradient force. Figure 2b shows that the $x$ component of the optical gradient force with laser power of 10 mW has a magnitude up to 3 pN at the upper edge of the nanohole sidewall. Such optical gradient force can keep the AuNU stably trapped at a 5-nm distance from the sidewall, because it overwhelms the Brownian motion as well as optical scattering force repelling the AuNU (Supplementary Note 1 and Supplementary Fig. 4)[7,21].

Furthermore, electrical bias is applied to suppress the AuNU motion in the z direction. As illustrated in Fig. 2a, when the surface charge of the AuNU and the nanohole are both negative, the AuNU in the nanohole under bias is exposed to both electrophoretic (EP) and electroosmotic (EO) forces[9]. The magnitude and direction of these two forces are proportional to the value and sign of the zeta potentials ($\zeta$) of the AuNU ($\zeta_{np}$) and the nanohole wall ($\zeta_{hole}$). As illustrated in Fig. 2c, d, the AuNU has an effective velocity in the z direction as follows[22]

$$v = \frac{\varepsilon E}{\eta}(\zeta_{np-}\zeta_{hole}) \qquad (1)$$

where $\varepsilon$ is the dielectric permittivity of the solution, $E$ is the z component of the applied electric field, and $\eta$ is the solution viscosity. The $\zeta_{np}$ of the AuNUs with multilayer adenines (A) attached on their surface (A–AuNU) is measured in the range between −18 and −20 mV (Supplementary Table 1), while the $\zeta_{hole}$ of alumina in 0.1 M NaCl at pH = 7.6 is around −20 mV[23]. As the zeta potentials of the A–AuNU and nanohole sidewall have the same sign (negative in our case), the EO force not only counterbalances the EP force but also suppresses the entropic

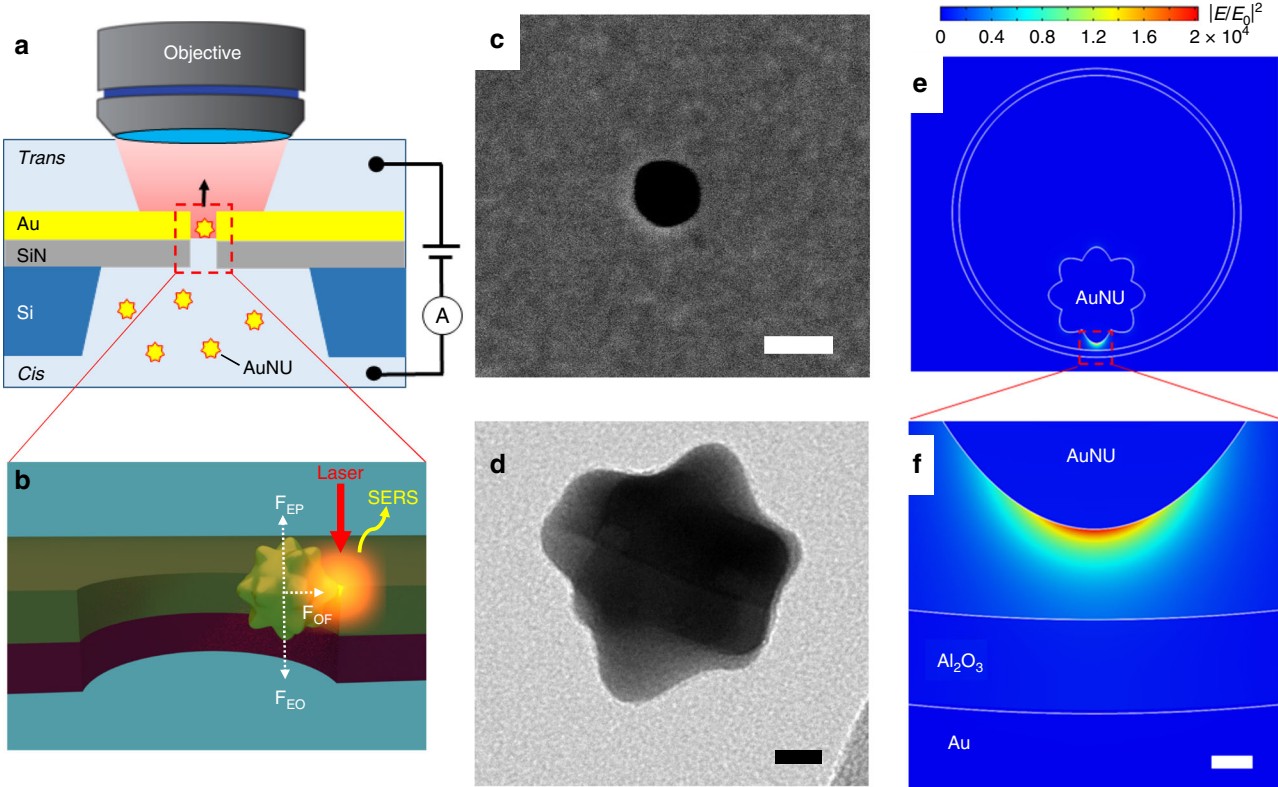

**Fig. 1** Electro-plasmonic trapping for single-molecule SERS. **a** Schematic of the flow-through setup that allows single gold nanourchin (AuNU) to flow through and be trapped under applied bias in a gold nanohole at plasmonic resonance upon the laser excitation at 785 nm. **b** The trapping due to a balance between the electrophoretic ($\mathbf{F}_{EP}$), electroosmotic ($\mathbf{F}_{EO}$), and optical ($\mathbf{F}_{OF}$) forces leads to a plasmonic hot spot between the AuNU tip and the nanohole sidewall that allows single-molecule SERS. **c** SEM image of the gold nanohole. The scale bar is 200 nm. **d** TEM image of the AuNU. The scale bar is 10 nm. **e** Simulated electromagnetic field intensity distributions of the AuNU coupled with the nanohole. The color bar represents the enhancement of the electromagnetic field intensity. **f** Magnified view of the intensity distribution of the electromagnetic field at one tip of the AuNU. The scale bar is 2 nm.

AuNU flow in the $z$ direction. The small difference between these two zeta potentials as well as the small amplitude of the applied electric potential reduces the velocity of the A–AuNU to a near-zero value in the nanohole. Therefore, in combination with the optical force, the resultant electroplasmonic trapping allows the AuNU to couple with the nanohole such that the AuNU tip exhibits a strong and highly confined electromagnetic field capable of single-molecule SERS.

**Controllable trapping for reproducible SERS.** The electro-plasmonic trapping of single AuNUs is controllable by turning on and off the bias. Among many A–AuNUs that flow through the nanohole, only those that are close to the sidewall can be trapped and detected, because of their strong SERS signals. A time trace of the 730 cm$^{-1}$ band intensity of the A–AuNU (red line in Fig. 3a) at 6-mW laser power and 1-V bias indicates that the A–AuNU is trapped in the nanohole until the bias is turned off. We notice that among four trapping periods, the third trapping period between 100 and 170 s exhibits higher peak intensity than those of the previous two periods. It could be due to trapping of a single AuNU with shorter gap distance between the AuNU tip and the nanohole wall, which provides larger field enhancement (Supplementary Fig. 3). Furthermore, another reason could be stronger electromagnetic coupling between the nanohole wall and another sharper tip of the same AuNU.

The stable and reproducible SERS spectra (Fig. 3b) suggest the trapping of a single A–AuNU rather than more than one A–AuNU. If more than one A–AuNUs are driven in the nanohole, many strong gap-based hot spots are generated due

to either inter-AuNU coupling or coupling between AuNU tips and the nanohole wall. However, the conformations of the adenine molecules adsorbed in these hot spots are not necessarily the same. Weak modes can be enhanced randomly by those hot spots to change the peak intensities and positions of the spectra, leading to strong fluctuation of both the peak positions and the baseline (Supplementary Fig. 5)[24]. When only one AuNU remains stably trapped in the nanohole, only one gap-based hot spot of the trapped AuNU tip near the nanohole sidewall remains and becomes dominant. Thus, the adenine on this tip surface is continuously excited to produce reproducible and stable spectra.

The trapping time of single AuNU is actually affected by an interplay between the AuNU hot spot and the trapping force. The AuNUs are not uniform in sizes and shapes, leading to slight difference in the distance between the AuNU tip and nanohole wall. This in return influences the hot spot strength, because smaller distance gives rise to a stronger hot spot and vice versa. When the AuNU is pulled close to the sidewall, the coupling between the nanohole and the AuNU tip becomes stronger to enhance further the optical force. As a result of this positive cycle of the optical force, some AuNUs may be pulled to even touch the nanohole wall temporarily, which can induce strong interactions to affect the SERS spectra as well as the trapping time[25,26]. With the feedback from the Raman spectra, the platform can be extended to automated trapping and high-throughput analysis of single AuNUs of interest with a closed-loop program[27].

The stable trapping can last for a few minutes and exhibits reproducible SERS signals with low relative standard deviation (RSD). For example, a RSD as low as ~13% is achieved by

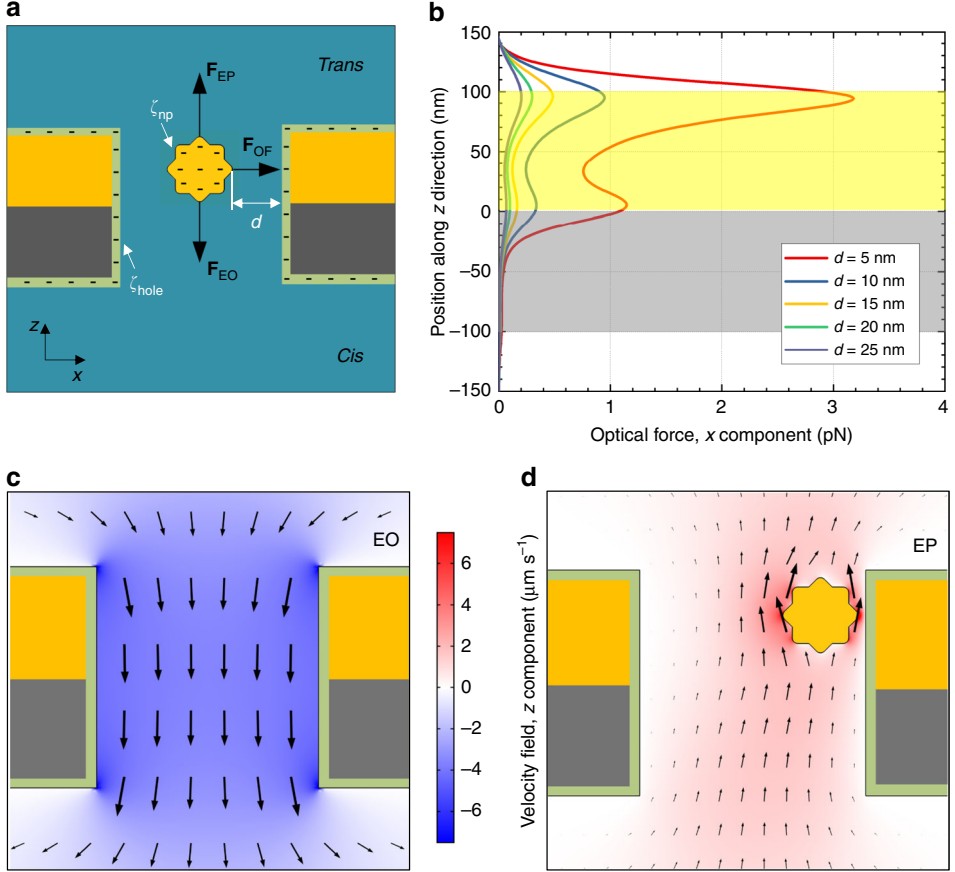

**Fig. 2 Electro-plasmonic trapping effect. a** Schematic illustration of the electro-plasmonic forces exerted on an AuNU in the nanohole under bias, both of which have negative surface charges. Black arrows represent the directions of electrophoretic ($\mathbf{F}_{EP}$), electroosmotic ($\mathbf{F}_{EO}$), and optical ($\mathbf{F}_{OF}$) forces. White arrows indicate the zeta potentials on the AuNU ($\zeta_{np}$) and on the nanohole wall ($\zeta_{hole}$), respectively, and $d$ is the distance between the particle tip and nanohole wall. **b** Simulated optical force under 10-mW laser of 785-nm wavelength for a varying particle position along $z$ direction with different $d$. **c** Simulated electroosmotic (EO) velocity field distribution, $z$ component. **d** Simulated electrophoretic (EP) velocity field distributions, $z$ component. Color bar represents the positive (red) and negative (blue) velocity in (**c, d**).

applying 1-V bias and 12-mW laser power (Supplementary Fig. 6), which is comparable to the RSDs of reproducible solid-state SERS substrates[24]. However, high laser power does not always lead to high signal reproducibility, because it also generates more heat (Supplementary Fig. 7), which increases the Brownian motion. Furthermore, since the $\zeta_{np}$ changed with different nucleotide molecules adsorbed on the AuNUs (Supplementary Table 1), electric potentials with different amplitudes of 1–4 V are used to ensure the stable trapping of AuNUs with different molecules (Supplementary Fig. 8). Tuning of both laser power and electrical bias is required to achieve single-molecule SERS.

**Single-molecule SERS of individual DNA bases**. We first use the bianalyte SERS technique (BiASERS) with cytosine (C) and isotope-edited cytosine ($C_{iso}$) molecules to demonstrate single-molecule capability of our platform[28,29]. In comparison with C, the $C_{iso}$ has some carbon and nitrogen atoms replaced by their isotopes (Supplementary Table 2). Importantly, both C and $C_{iso}$ exhibit the same surface area (1.27 nm²), Raman cross section, and affinity to gold surface. Due to the isotope atoms, the SERS peak of $C_{iso}$ at around 941 cm⁻¹ is shifted from that of C at around 904 cm⁻¹ (Supplementary Fig. 8). To demonstrate single-molecule SERS, submonolayers of equal moles of C and $C_{iso}$ molecules are adsorbed on the AuNU ($C_{iso}$C–AuNU) with a

surface coverage around one molecule per 4 nm². As shown in Fig. 4a, b, one SERS peak appearing at the C range (900–930 cm⁻¹) represents detection of a single C molecule, and the emergence of one SERS peak at the $C_{iso}$ range (930–960 cm⁻¹) represents the detection event of a single $C_{iso}$ molecule. If both peaks appeared, they belong to a multi-molecule event.

The BiASERS analysis requires >1000 events (SERS spectra) to statistically prove single-molecule SERS (see Methods for data-processing details), which actually depends on a sensitive and localized hot spot on the AuNU tip. If the hot spot has a small size to cover only one molecule, the probabilities of detecting either one of the molecules are high. If the hot spot became large enough to cover >1 molecule, the probability of detecting both of them increases as well. When the former is larger than the latter in our case of around 1030 events (Fig. 4a), a BiASERS histogram with more single-molecule events than multi-molecule events (Fig. 4c) is established to confirm single-molecule detection.

To prove the validity of the approach, we repeat the BiASERS experiments for single-molecule detection of all four DNA bases, in which submonolayer combinations of two of the four DNA bases are attached on the AuNUs (see Methods for details)[4]. We mix equal moles of adenine and guanine (G) with the AuNU solution to allow an average submonolayer coverage of around one molecule per 2 nm² on the AuNU surface (AG–AuNU). Around 1200 SERS data (Fig. 4d, e) are processed to produce the BiASERS histogram of the AG–AuNUs (Fig. 4f), proving again

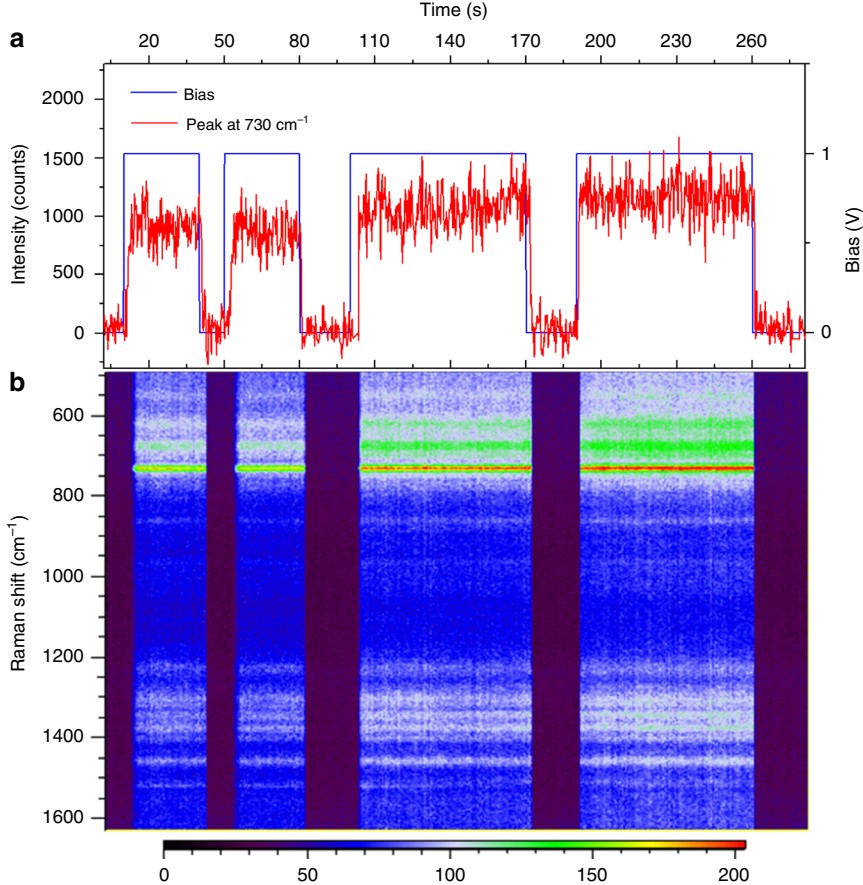

**Fig. 3** Reproducible SERS spectra of a trapped adenine-multilayer-adsorbed gold nanourchin (A–AuNU) by controllable trapping through turning on and off the bias. **a** Time trace of intensity of the Raman peak at 730 cm$^{-1}$ (red line) of a single A–AuNU trapped in a nanopore by incident laser of 6-mW power and applied bias of 1 V (blue line). **b** Corresponding time series of the stable SERS spectra produced by the trapped A–AuNU. The color bar represents the intensity.

single-molecule detection. The fact that more single-G events than single-A events is due to different nucleobase affinities to gold surface (A > C ≥ G > T)[30] as well as different trapping times of the AuNUs. For example, the AG–AuNU 2 exhibiting only one peak at the G range (660–700 cm$^{-1}$) is trapped longer than the AG–NU 3 that exhibited only one peak at the A range (700–740 cm$^{-1}$).

The single-molecule spectra from different AG–AuNUs in Fig. 4d exhibit considerable peak shifts of up to 15 cm$^{-1}$ in comparison with SERS spectra of a multilayer nucleobase without electrical bias in literatures[31,32]. For example, the guanine peaks at around 675 cm$^{-1}$ of the AG–AuNU 2 are red shifted from the multilayer peak at 660 cm$^{-1}$ (black dotted line). This is also seen for the adenine peaks at around 715 cm$^{-1}$ of the AG–AuNU 3 that are shifted from the multilayer peak at 730 cm$^{-1}$ (black dotted line). Interestingly, peak shifts up to 15 cm$^{-1}$ were also observed in a tip-enhanced Raman spectroscopy (TERS) experiment that used a gold nanotip to press an adenine nanocrystal[33]. The TERS peak shift was attributed to molecular reorientation during drifting of the gold nanotip that deformed the pressurized adenine molecule[34,35].

When the AuNU-sidewall distance is short enough, the AuNU tip mimics the gold nanotip moving close to a gold film in TERS experiments. Accordingly, the following five effects could be induced to change the SERS spectra. First, an effect of local field gradient is possible that the local electromagnetic field can change intensity greatly over a few angstrom[36]. The field gradient effect could even induce electric dipole–electric quadrupole

polarizability tensor of the adsorbed molecules to activate Raman-inactive bands[37,38]. Second, the increased plasmonic force at the AuNU tip due to the increased field gradient could pull the adsorbed molecule[3]. Third, the local temperature at the AuNU tip could increase to allow fast molecule motions[39], including rotation and diffusion on the AuNU tip[40,41]. Fourth, the adsorbed molecule can be deformed by the interaction between the electric double layers on the AuNU tip and the alumina layer on the nanohole wall, respectively. Finally, an oscillation of the trapped AuNU in the nanohole may induce lateral drifting of the AuNU tip. If the adsorbed molecule is pressed by the electric double layers, the drifting may change its orientation on the AuNU tip.

The universal shifting with similar amplitudes for both single-molecule adenine and guanine peaks can be due to molecule reorientation on the AuNU surface by the applied electric field that is used to trap the AG–AuNUs[42–45]. Under a static electric field, nonsymmetrical molecules had field-induced dipole moments and were reoriented by the electric field, exhibiting significant SERS peak shifts[42,45]. In our case, both adenine and guanine molecules are nonsymmetrical molecules, because they both have one six-membered ring and one five-membered ring. Therefore, they may be easily reoriented by the applied electric potential (4 V), leading to strong peak shifts.

Furthermore, the peaks of the AG–AuNU 2 and AG–AuNU 3 that shift to opposite directions can be ascribed to different initial states of orientations on the AuNU tip. For example, rotations of nonsymmetrical molecules on a gold film at 90 K were observed

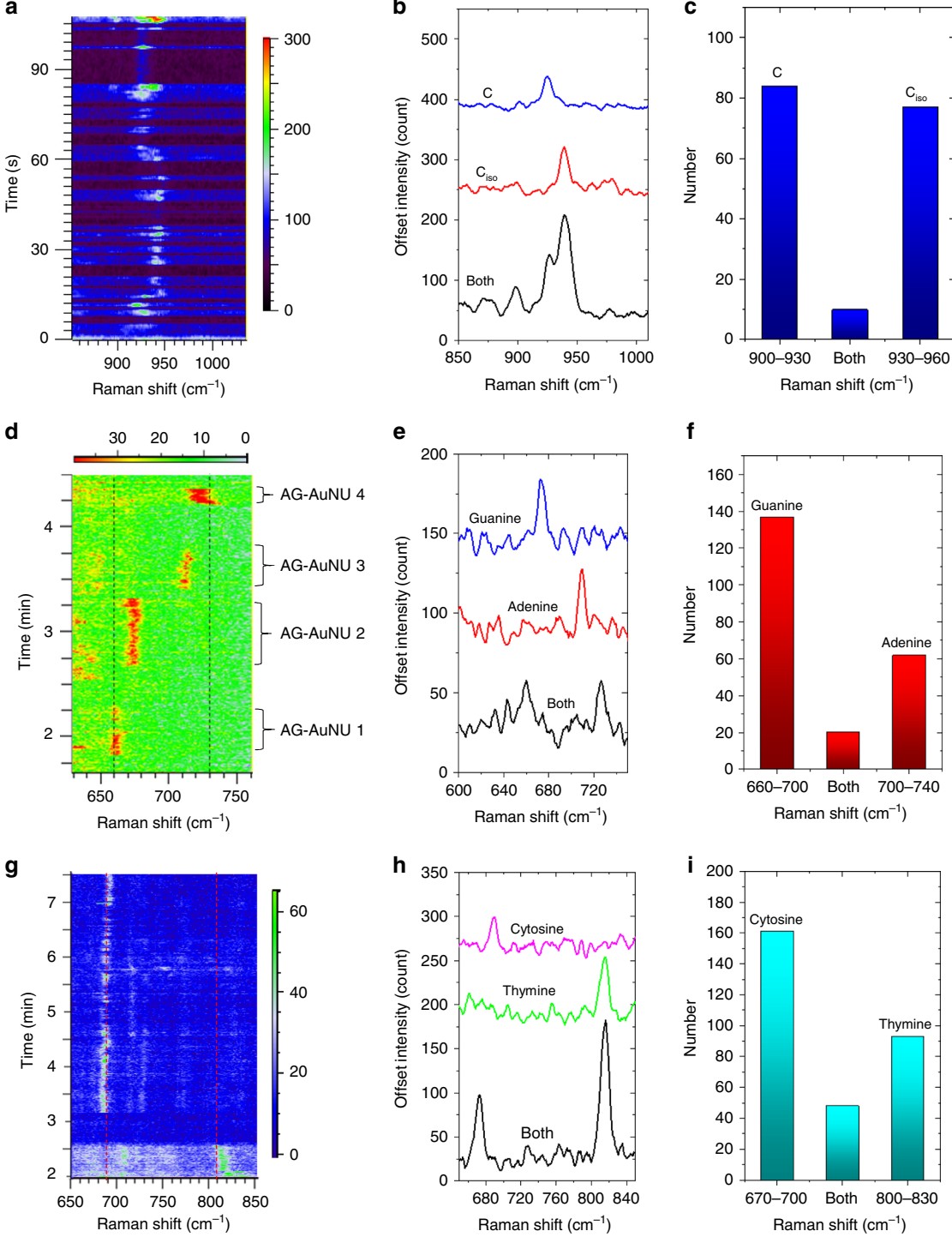

**Fig. 4** BiASERS analysis of single-molecule SERS of individual nucleobases. **a** SERS time series of the gold nanourchins with submonolayers of equal moles of cytosine (C) and isotope-edited cytosine ($C_{iso}$) molecules adsorbed on their surface ($C_{iso}C$–AuNUs) trapped in gold nanoholes. **b** Representative single-molecule spectra (colored curves) and multi-molecule spectrum (black curve) of $C_{iso}C$–AuNU, respectively. **c** BiASERS distribution of $C_{iso}$ and C out of about 1030 spectra. **d** SERS time series of the gold nanourchins with submonolayers of equal moles of adenine (A) and guanine (G) molecules adsorbed on their surface (AG–AuNUs) trapped in a nanohole. **e** Typical SERS spectra of single-molecule spectra (colored curves) and multi-molecule spectrum (black curve) of AG–AuNU. **f** BiASERS distribution of A and G out of about 1200 spectra. **g** SERS time series of the gold nanourchins with submonolayers of equal moles of C and thymine (T) molecules adsorbed on their surface (CT–AuNUs) trapped in a nanohole. **h** Typical SERS spectra of single-molecule spectra (colored curves) and multi-molecule spectrum (black curve) of CT–AuNU. **i** BiASERS distribution of C and T from about 1500 collected spectra. All color bars represent the signal-to-baseline intensity of the SERS peaks. The dotted lines indicate SERS peak positions of multilayer nucleobases (black for A or G, red for C or T) adsorbed on silver nanoparticles without bias, respectively, as reported in ref. [31].

to shift Raman peaks in opposite directions due to reversible orientation states. Such molecular reorientations were induced by the ambient thermal energy and could happen faster at room temperature[40]. Similarly, the A and G molecules on the AuNU at different initial orientation states in the more energetic environment of the electroplasmonic trap can give rise to the opposite peak shifting.

Similar single-molecule detection of the cytosine and thymine (T) is achieved by mixing equal moles of the two bases with the AuNU solution to form submonolayer CT–AuNUs (Fig. 4g). Yet, the surface areas of the C (1.27 nm$^2$) and T (1.42 nm$^2$) are smaller than those of A and G. Furthermore, C adsorbs on gold surface with an angle tilted from the surface in solution phase[46]. As a result, many C and T molecules are covered in a hot spot on the AuNU tip that produces higher peak intensity, such as the "Both" spectrum in Fig. 4h. Therefore, the BiASERS distribution of the CT–AuNUs (Fig. 4i) is not achieved until the CT coverage on the AuNU surface is decreased to one molecule per 7 nm$^2$ in order to lower the probabilities of including >1 molecule in the hot spot. Long trapping is observed as the cytosine peak intensity at around 685 cm$^{-1}$ fluctuates from 3.1 to 7.5 min under a 1- V bias and a 6-mW laser power (Fig. 4g), leading to more single-C events than single-T ones. In comparison with the case of AG–AuNUs, the single-molecule SERS peaks of the cytosine and thymine shift with a smaller amplitude (Fig. 4g). The small shift amplitude can be due to the small bias (1 V) as well as symmetrical molecule structures of cytosine and thymine that both have only one six-membered ring.

We notice that BiASERS detections of single DNA bases by metal nanoparticles either in solutions or on solid-state wafers were not reported before. One reason was due to the small Raman

scattering cross section of the DNA bases. For example, the Raman cross section of adenine was 10–100 times less than dyes[47]. Besides, in common BiASERS detection of >1000 SERS spectra by, for example, silver colloids, colloid clusters were instantly formed during their diffusion in solutions[4]. Such >1000 SERS spectra came probably from similar amount of different colloid clusters that had different field enhancement, hot spot size, and molecule adsorption in the hot spots. In strong contrast, the >1000 SERS spectra for the BiASERS analysis in our method come from a few single AuNUs trapped in our platform for tens of seconds, respectively. The resultant hot spots are formed after the molecule adsorption to ensure that the same molecule is excited reproducibly by the same intense hot spot with similar size.

**Discrimination of single nucleobases in a DNA.** To demonstrate the discrimination of a single nucleobase in a DNA strand, submonolayers of oligonucleotides of 5′-CCC CCC CCC A-3′ (9C1A), 5′-C AAA AAA AAA-3′ (1C9A), and 5′-AAA AAA AAA CTG-3′ (9ACTG) are directly adsorbed on the AuNUs, respectively (see Methods), before the AuNUs were driven to the nanoholes for detection. Unlike a normal helix structure in solution, oligonucleotide molecules are adsorbed by nonspecific binding of the nucleobases on the gold surface[48,49]. The SERS scattering cross sections of the nonspecific binding nucleobases of a oligonucleotide were found to be A≈C > G, T[50]. Due to specific surface selection rules, nucleotides with different conformations on the gold surface exhibit SERS spectra (Fig. 5) different from individual DNA bases[51,52]. Therefore, information of the oligonucleotide conformation can be learnt from the SERS spectra of single nucleobases of the oligonucleotide[50,53].

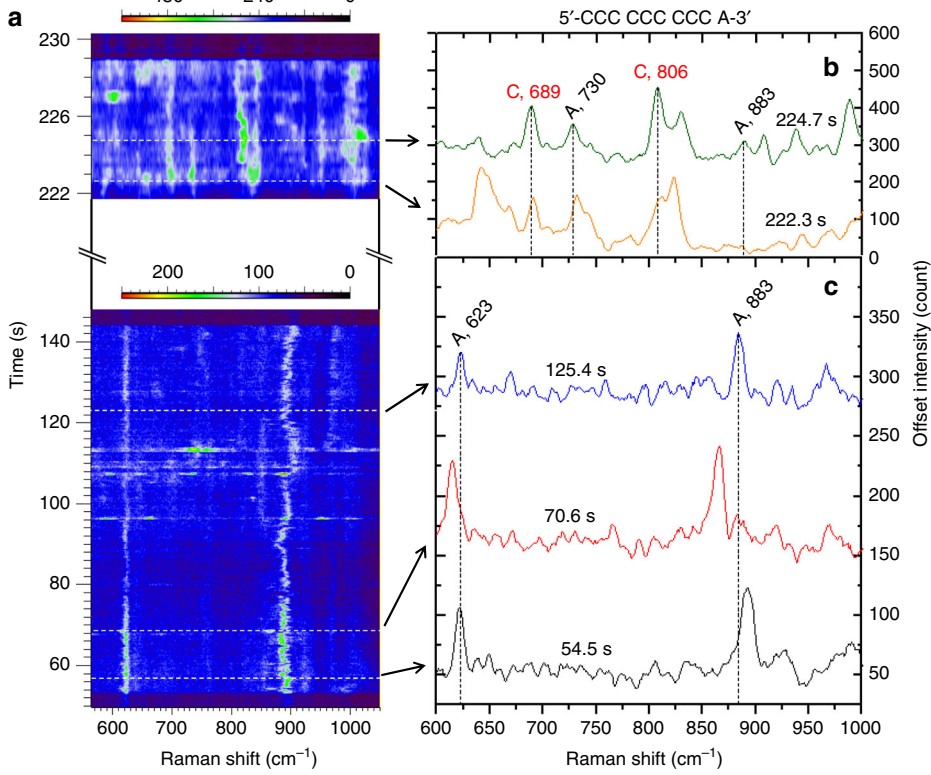

**Fig. 5** SERS discrimination of single nucleobases in single oligonucleotides. **a** SERS time series extracted from 1400 spectra produced by adsorbing 5′-CCC CCC CCC A-3′ oligonucleotide submonolayer on the gold nanourchins (9C1A–AuNUs) and trapping them in the nanohole. The two color bars represent the signal-to-baseline intensity of the Raman modes in different time spans, respectively. The white dotted lines and black arrows indicate **b** the mixed SERS spectra of both A and C, and **c** the single-A spectra, at specific times. The black dotted lines in (**b**, **c**) indicate the frequency positions of the Raman peaks unique to A and C, respectively[31,32].

In the single-A spectra of the 9C1A oligonucleotide (Fig. 5a, c), the peaks at around 623 and 883 cm$^{-1}$ are assigned to six-ring and five-ring deformations, respectively[31]. Their spectral wandering is a typical signature of single-molecule SERS[54,55]. However, the strong ring breadth mode of adenine at around 730 cm$^{-1}$ is not present in most single-A spectra (Fig. 5a) until mixed spectra of both adenine and cytosine are detected at 222.3 s (Fig. 5b). The absence of the ring breadth mode represents a parallel geometry of the adenine molecule on the gold surface[56,57], which agrees with previous reports[58,59]. Therefore, it suggests that the single adenine is identified only when the 9C1A molecule is stretched on the AuNU surface presumably by the electro-plasmonic forces[3].

Identification of other single nucleobases is demonstrated in the cases of 1C9A (long red arrows in Supplementary Fig. 9) and 9ACTG (long colored arrows in Supplementary Fig. 10) oligonucleotides. Among so many SERS spectra of the oligonucleotides, only small parts of them demonstrate single nucleobases: about 6.25% single-A data in the 9C1A spectra and only 0.16% single-C data in the 1C9A spectra, respectively. In the case of 9ACTG, there are 1.41% single-C, 2.10% single-T, and 4.75% single-G data, respectively. The detection probabilities of each nucleobase agree partially with the nucleobase affinities to gold surface $(A > C \geq G > T)$[30], and the rest depends mainly on the different trapping times of the AuNUs.

## Discussion

To conclude, by illuminating a plasmonic resonant nanohole, we are able to trap a AuNU in the nanohole on demand by applying an electrical bias. Plasmonic hot spots are formed by coupling the AuNU tip and the nanohole for reproducible SERS detection of nucleotides that are adsorbed on the AuNU before the trapping. The trapping is so controllable that the AuNUs with SERS signals of interest to us are trapped for minutes. As a result, the nucleotides stay in the hot spots sufficiently long such that single-molecule detections of all four DNA bases and single-nucleobase discrimination in oligonucleotides are demonstrated.

Our method utilizes direct adsorption of molecules on the AuNU rather than thiol chemical bonding, which is universal to all molecules. By measuring the zeta potential of the molecule-adsorbed AuNUs, we can coat the nanohole with an oxide layer with a similar zeta potential for particle trapping and detection. Accordingly, our platform can be widely applied to reliable SERS sensing of single molecules such as amino acids[47,60]. Meanwhile, structural and conformational information of proteins has been obtained by analyzing SERS spectra from amino acids in protein subdomains[61–63]. By trapping gold nanospheres, our method can also obtain hot spots of different sizes to cover one or more amino acid residues of a protein molecule. Therefore, our method can also be a promising SERS platform to detect amino acids and proteins.

## Methods

**Materials**. Nonfunctionalized gold nanourchins (AuNUs) with average particle size of 50 nm and concentration of $3.5 \times 10^{10}$ particles/mL were obtained from Sigma (795380-25 ML). The nucleobases adenine (A), cytosine (C), isotope-edited cytosine (C$_{iso}$), guanine (G), and thymine (T) were obtained from Sigma: A (A8626), C (C3506), C$_{iso}$ (492108), G (G11950), and T (T0376). Phosphate Buffered Saline (806552 Sigma) was used for preparation of the samples and Raman measurements.

**Attachment of multilayer DNA bases on the AuNUs**. The DNA nucleobase molecules were adsorbed on the surface of AuNUs in a condition that still kept the AuNUs stable in the electrolyte. To obtain the stable nucleobase-AuNUs, the concentration of AuNUs and the salt concentration were $1.3 \times 10^{10}$ particles per mL of AuNUs and 1.25% of phosphate buffer. The multilayer nucleobase–AuNUs were prepared by pipetting 300 μL of the AuNU suspension from a stock in an Eppendorf tube, which was then resuspended in 400 μL of deionized water. Nucleobase solution with a final concentration of 125 μM was

### Table 1 Oligonucleotide concentrations needed to form monolayers on the AuNUs.

|  | Oligonucleotide surface area (nm²) | Number of oligonucleotides per AuNU | Oligonucleotide (mole) per $1.3 \times 10^{10}$ mL$^{-1}$ AuNUs |
|---|---|---|---|
| 1C9A | 14.05 | 559 | $9.28 \times 10^{-12}$ |
| 9C1A | 12.85 | 611 | $1.01 \times 10^{-11}$ |
| 9ACTG | 17.01 | 461 | $7.66 \times 10^{-12}$ |

added in the AuNU solution and allowed to adsorb on the AuNUs with continuous mixing on a shaker at room temperature for 3 h prior to the Raman measurement. The suspension of the AuNUs with a nucleobase was stable in a refrigerator at 4 °C for up to 2 weeks.

**Sub-monolayer attachment of AG, CT, and C$_{iso}$C on the AuNUs**. For the BiASERS experiments, submonolayers of nucleobase were adsorbed on the surface of the AuNUs where lower concentrations of nucleobase solution were used, according to the surface area of nucleobases on gold surface, i.e., A (1.42 nm²), G (1.54 nm²), C and C$_{iso}$ (1.27 nm²), and T (1.42 nm²). To obtain the stable nucleobase-AuNUs, the concentration of AuNUs and the salt concentration were $1.3 \times 10^{10}$ particles per mL of AuNUs and 1.25% of phosphate buffer. Then, the average number of nucleobases adsorbed on one AuNU were controlled at 5000 for the BiASERS experiment of AG detection. That is, 50 nM final concentration of each nucleobase solution of adenine and guanine was added to the AuNU solution. Similarly, 10 nM of each nucleobase solution of cytosine and thymine were added for BiASERS experiments of the CT detection with 1000 nucleobases per AuNU on average. For the BiASERS experiments with isotope cytosine and cytosine, 21.6 nM of isotope cytosine and cytosine solutions were added to achieve 2000 nucleobases (1000 C$_{iso}$ + 1000 C) per AuNU on average. The incubation time to form a sub-monolayer of adsorbed mixture of nucleobase on the AuNUs was 24 h prior to the Raman measurement.

**Sub-monolayer attachment of oligonucleotides on the AuNUs**. The concentration of oligonucleotides required to achieve monolayers on the AuNU surface was determined, considering the surface area of nucleobases on gold surface and calculating the area occupied by a single oligonucleotide as shown in Table 1. The surface area of a single φ50-nm AuNU is calculated as 7850 nm² by regarding it as a φ50-nm gold nanosphere. Knowing the amount of oligonucleotide molecules required to form a monolayer on a single AuNU, we then calculate the μM of oligonucleotide needed to form a monolayer on AuNUs with a concentration of $1.3 \times 10^{10}$ mL$^{-1}$.

In detail, 300 μL of AuNU stock solution was dispersed in 400 μL of 5% PBS, pH 5.5, to make $1.3 \times 10^{10}$ mL$^{-1}$ AuNUs. An aliquot of 100 μL of oligonucleotide solution in the same buffer was added to reach the desired concentration for the monolayer formation (final volume 800 μL): 11.6, 12.68, and 9.58 nM, respectively, for 9A1C, 1A9C, and 9ACTG. After vortexing, the samples were left at 4 °C for 2 days allowing the spontaneous absorption of the oligonucleotide molecules on the AuNU surface. Due to the different conformation and adsorption of the oligonucleotide on the nonuniform AuNUs, the oligonucleotides actually formed a submonolayer on the AuNUs.

**Measurement of dynamic light scattering and absorbance of the colloid**. Dynamic light scattering experiments were performed by using a Malvern Zetasizer, and the measurements were evaluated by using Zetasizer software. Data are reported as the average of three measurements. Particle diameter, PDI, and zeta potential were used to characterize the colloid suspension before and after functionalization and to evaluate its stability over time. Unless otherwise mentioned, particles were analyzed at a diluted concentration of $1.3 \times 10^9$ particles per mL in filtered deionized water and in the buffer solution used for their synthesis at 25 °C in disposable folded capillary cells (DTS1070). By using aqueous solutions as dispersants at 25 °C, 0.8872cP and 78.5 were used as parameters for temperature, density and dielectric constant, respectively, during the measurements. RI 0.2 and an absorption of 3.32 were used as parameters for the analysis of gold nanomaterials. Cary300 UV–Vis (Varient Aligent) was used for the UV–Vis absorption analysis of the colloidal suspension. The absorbance spectra were recorded by using samples at diluted concentration of $6.5 \times 10^9$ particles per mL in water or in the buffer used for the synthesis. Absorbance measurements were performed by using a 300–800-nm wavelength range in a disposable plastic cuvette (1-mL maximum volume, 1-cm path distance).

**Fabrication of the nanohole devices and PDMS encapsulation**. After sputtering a 2-nm-thick titanium and 100-nm-thick gold layer on the front side of the SiN

membrane, as well as a 2-nm-thick titanium and 20-nm-thick gold layer on its back side, focused ion beam milling (FEI Helios NanoLab 650 DualBeam) at a voltage of 30 keV and a current from 0.23 to 2.5 nA was used to drill hole arrays in the back of the Ti/Au-coated SiN sample. Then, an alumina layer of 5 nm was deposited on the back of the sample by atomic layer deposition (Oxford Instruments). The sample was annealed on a hot plate at 200 °C in air for 1 h and allowed to cool naturally. The as-made nanoholes were embedded in a microfluidic chamber made from polydimethylsiloxane (PDMS, Dow Corning SYLGARD 184 silicone elastomer) cured at 65 °C for ~40 min.

**Simulations**. COMSOL Multiphysics® software was used to perform the modeling of the optical trapping force, electroosmotic, and electrophoretic flows (Ref. COMSOL Multiphysics® v. 5.3a. www.comsol.com. COMSOL AB, Stockholm, Sweden). 3D models were constructed to solve the optical trapping, electromagnetic heating, and fluid dynamics problems. The optical trapping force was obtained by integrating Maxwell's stress tensor over the particle surface. Perfectly matched layers were used over the whole domain to prevent backscatter from the boundaries. A Gaussian beam was illuminated from the top side. The refractive index and thermal conductivity of gold, silicon, and aluminum oxides were taken from the COMSOL library, and the refractive index of the fluid was set to 1.33. The electroosmotic and electrophoretic flows were obtained by solving the Poisson–Nernst–Planck equations. The electrical conductivity, relative permittivity, density, and dynamic viscosity of the fluid were 1.5 S m$^{-1}$, 79, 1000 kg m$^{-3}$, and 0.8 mPa s, respectively. The ζ of the hole and particle surfaces were –20 mV and no-slip boundary conditions were applied to the remaining surfaces.

**Raman measurements**. Raman measurements were conducted by a Renishaw inVia Raman spectrometer with a Nikon 60× water immersion objective with a 1.0 N.A., a 785-nm laser, and an exposure time of 0.1 s. The laser beam was focused to a spot diameter of 1.5 μm with a power varying from 2 to 12 mW.

**Single-molecule data processing**. Spectra were processed by using custom python scripts according to Chen et al.[2] Data were selected in a 500–1000 cm$^{-1}$ window and smoothed by using a Savitzky–Golay filter. A baseline was fitted to each spectrum by using fifth-order polynomial functions and subsequently removed from the spectra. Peaks were detected on the resulting spectra by using pythons' signal_find_peaks_cwt function. The final peaks were selected if the peak height exceeds four standard deviations of the spectra for the C$_{iso}$C–AuNU data and two standard deviations for the AG–AuNU and CT–AuNU data, respectively.

**Reporting summary**. Further information on research design is available in the Nature Research Reporting Summary linked to this article.

## Data availability
The authors declare that all the data supporting the findings of this study are available within this paper and its Supplementary Information file, or available from the corresponding author upon reasonable request.

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

## Acknowledgements
The authors thank Dr. Michele Dipalo and Dr. Daniel Darvill for drawing of schematic figures and editing, respectively. The research leading to these results has received funding from the Horizon 2020 Program, FET-Open: PROSEQO, Grant agreement no. [687089].

## Author contributions
J.A.H. designed the experiment, carried out Raman experiments, and wrote the manuscript. M.Z.M., Y.Z., and G.G. handled the molecules' attachment on the AuNUs and buffer optimization for the functionalized AuNU stability. G.G. and Y.Z. measured the zeta potentials and absorbance of the functionalized AuNUs. Y.Z. fabricated the nanohole device. A.H. and F.O. did the simulation and investigated the trapping mechanism. J.A.H., A.H., and M.S. analyzed the data. F.D.A. and D.G. supervised the work. All authors discussed the results and contributed to the final paper preparation.

## Competing interests
M.S. is an employee of Alacris Theranostics GmbH. The remaining authors declare no competing interests.

## Additional information

**Peer Review Information** *Nature Communications* thanks the anonymous reviewers for their contribution to the peer review of this work. Peer reviewer reports are available.

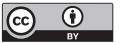

