## [Peer Review File · Nature Communications]

Reviewers' comments:

Reviewer #1 (Remarks to the Author):

The authors present an interesting report on SERS in plasmonic nanoholes, enabled by trapping of Au nanoparticles in the holes. They demonstrate long trapping times and show a convincing equilibrium of optical forces, electro-phoretic and electro-osmotic forces and brownian dynamics. This indeed likely results in single molecule SERS. The evidence for this, however, is not convincing in the same way. The authors claim they used BiasSERS, which is partially true, with the caveat that the used molecules are chemically different, i.e. they have a different affinity to the Au surface, such that it is not obvious that seeing only one spectrum really means only one molecule is present. (usually isotopes of the same molecule are used to circumvent this). Also they use the argument that the orientation is resulting in shifted Raman peak positions, to plead for single molecule characteristics. It has been shown before in single particle SERS (also in plasmonic nanoholes - Kerman et al, *Nanoscale*, 2015,7, 18612), that this type of effects occur, and are likely related to temporal adsorption to the the gold surface, and the resulting stress that leads to the peak shifts (this can be the case too for monolayers).

Overall: the results show interesting indications, but not real proof yet for single molecule sensitivity.

In addition, they could indeed be used for SERS detection of small molecules, but the general applicability for a broader set of markers is questionable.

Reviewer #2 (Remarks to the Author):

The authors propose a plasmonic nanohole assisted trapping system to detect SERS signal of DNA bases at a single molecular level. Gold nano-stars could be trapped by the combined effect of electroosmotic, electrophoretic, and optical forces. Due to hot spots excited in the gap between the AuNU and hole sidewall, it is able to distinguish different molecules with reliable and stable results. However, the Raman signals in this work seem stochastic rather than controllable when multiple samples are mixed, and there have been many approaches to get a simple single molecular Raman spectrum in the plasmonic tweezers.

My major concerns:

1. It has widely been demonstrated that hot spots could also be excited at the tip of Au nanourchins even without the nanoholes (*The Journal of Physical Chemistry C*, 2008, 112(48): 18849-18859; *Applied Physics Letters*, 2009, 94(15): 153113; *J. Am. Chem. Soc.* 2015, 137, 33, 10460-10463; etc.). As a result, the authors should provide comparable results without the plasmonic nanohole, to highlight its key role of the hole plays in this system.

2. In Fig.4, the Adenine and Guanine are mixed with equal moles (also Cytosine and Thymine), why the measured number of G/C is almost twice of A/T? And, the concentration of C-T is much lower than A-G, why the detected C-T's peak intensity is much higher?
3. In Fig. 4a, there exhibits considerable peak shifts for AG-AuNUs, it makes sense to attribute to molecule reorientation on the AuNU surface. But, how to explain the two peaks shifting in opposite directions?
4. The author mentioned that there will be 'a strong fluctuation of both the peak positions and the baseline' when there are multiple particles trapped, I wondered if the fluctuations of the peak and baseline are synchronous, as I find that there is an obvious mismatch at the time of ~670s and ~850s in Figure S5.
5. For the sample 9C1A in Fig.5a, I suppose the Raman peak of Cytosine will be much easier to be detected as there are more C than A. But I find a reversed result. The authors gave out an explanation that the affinity of A was the biggest in the four bases. If this is the right reason, for the sample of 1C9A, I think, it will be more difficult to detect the signal of Cytosine. However, in Figure S8, I find that the Cytosine's signal seems even easier to be detected compared with Adenine. For this, I think the discussion is not strong enough to support such a conclusion.

Given this, I cannot recommend its publication in Nature Communications.

Reviewer #3 (Remarks to the Author):

The manuscript by Huang et al demonstrated single molecule SERS detection of DNA bases in oligonucleotides. The detection method was based on trapping single gold nanourchin (AuNU) into a plasmonic nanohole to form a greatly enhanced electromagnetic field, which facilitates the detection of biomolecules of small Raman Scattering cross sections. The confinement of the AuNU inside the nanohole was achieved through long-time electro-plasmonic trapping, which is desirable for flow-through SERS detection. While the manuscript is interesting and the concept advances the field, the authors should consider performing a few more experiments to make a compelling case.

1. In Figure 3b, the author showed the contour map of the Raman spectra produced by a trapped A-AuNU. The on and off presence of the Raman signal is achieved through turning on and off the bias. However, a significant increase in the peak intensity is noticed consistently for the third and fourth ON periods. Is it possible that more AuNUs are getting trapped during those periods?
2. In the caption of Figure 1f, the authors indicate that it is magnetic field distribution. Why did the authors choose to the magnetic field instead of electric field? If it is magnetic field, the authors have to include a color scale bar and discuss the significance of the magnetic field distribution in this context.

3. It would be better to include more information related to the zeta potential of the AuNU before and after modifying various analytes. Since zeta potential plays a critical role for colloidal stability and electro-plasmonic trapping, it might be useful for the reader know the values.
4. The author has mentioned that all the analytes are absorbed on the AuNU through physical absorption. Hence, more experimental evidence of the efficient co-absorption and co-presence of analytes (such as Adenine and Guanine) on the AuNU is needed to further support the SERS spectra obtained from single-molecule measurement. The electromagnetic field intensity achieved by trapping the AuNU is within the range that can be achieved normal electromagnetic hotspots formed between assemblies of nanoparticles. Maybe the authors can harness such hotspots to further understand the adsorbate composition.
5. We suggest SERS measurement of AuNU alone as a negative control.
6. Also, based on all the experiments performed, which serve as training set, it would be great to design and implement a few double-blinded studies to demonstrate that the SERS spectra can provide accurate determination of the DNA bases in unknown oligos. This will make a compelling case that the demonstrated approach is indeed powerful for the detection of single bases in DNA.

Reply to referees

Thank you for your comments that helps to improve our manuscript. The manuscript has been intensively revised. In most cases, we accepted the suggestions and made the changes accordingly. In a few cases, we maintained the original version for reasons reported point-by-point in the following. The details of our replies are reported in blue following Reviewers' comments as below:

Reviewer #1 (Remarks to the Author):

The authors present an interesting report on SERS in plasmonic nanoholes, enabled by trapping of Au nanoparticles In the holes. They demonstrate long trapping times and show a convincing equilibrium of optical forces, electro-phoretic and electro-osmotic forces and brownian dynamics. This indeed likely results in single molecule SERS. The evidence for this, however, is not convincing in the same way. The authors claim they used BiaSERS, which is partially true, with the caveat that the used molecules are chemically different, i.e. they have a different affinity to the Au surface, such that it is not obvious that seeing only one spectrum really means only one molecule is present. (usually isotopes of the same molecule are used to circumvent this).

Reply: we thank the Reviewer for the helpful suggestions. In light of provided advices, we carried out additional experiments. In particular, we performed BiASERS analysis with cytosine (C) and isotope-edited cytosine (C_{iso}) molecules submonolayer adsorbed on the gold nanourchins ($C_{iso}C$ -AuNUs). Both C and C_{iso} have the smallest area (1.27 nm^2) among the 4 nucleobases and 6181 Cytosines are needed to form a monolayer on a AuNU of 50 nm in diameter. Therefore, we designed the experiment by allowing adsorption of 2000 molecules at maximum ($1000 C_{iso} + 1000 C$) on one AuNU, which was approximately 1 molecule per 4 nm^2 . The SERS mode of the multilayer C_{iso} -AuNU at around 950 cm^{-1} is shifted from that of multilayer C-AuNU at around 904 cm^{-1} due to the isotope atoms, as shown in Figure R1a below. This mode was chosen for our BiASERS analysis because it has high intensity at the concentration level of submonolayer without neighbouring peaks to affect fitting its position. In the SERS spectra of the trapped $C_{iso}C$ -AuNUs, the numbers of peaks at either the C range ($900 - 930 \text{ cm}^{-1}$ representing only C) or the C_{iso} range ($930 - 960 \text{ cm}^{-1}$ representing only C_{iso}) are much larger than the number of 2 peaks appearing at both ranges. Such BiASERS distribution (in Figure R1d below) proved single-molecule detection by our particle trapping method.

While the BiASERS distribution with the C and C_{iso} proved single-molecule sensitivity of the trapped $C_{iso}C$ -AuNUs, the BiASERS distributions based on submonolayer of mixed nucleobases (AG-AuNU and CT-AuNU) were even stronger evidence of single-molecule SERS. Indeed, the different nucleobase affinities to gold surface ($A > C \geq G > T$)¹ can lead to adsorption of, for example, more Adenine (A) molecules than Guanines (G) on the AuNU tip in the case of submonolayer AG-AuNU. Nevertheless, a hot spot at the AuNU tip with a size similar to either molecules becomes critical. If the hot spot covered 2 A molecules and

exhibited 1 peak at the A range, it could also cover 1 A and 1 G molecules because they have similar sizes (A: 1.42 nm², G: 1.54 nm²). Consequently, the number of spectra with 2 peaks at both A and G ranges will be more than the number of 1 peak at the G range. In the extreme case that all AuNU tips were occupied by A molecules, the numbers of 1 peak at the G range and 2 peaks at both A and G ranges will be zero. But neither of them are the case in our BiASERS distributions for both AG-AuNU and CT-AuNU. In fact, in Le Ru *et al.*'s seminal paper of the BiASERS method,² statistics of more than 1000 spectra of 2 different analyte molecules was developed to remove the possibility that one Raman peak at the spectral range of either one of the 2 analyte molecules represents >1 molecules.³ However, the use of 2 different analyte molecules was difficult to achieve the BiASERS distributions experimentally in some cases, due to different affinities and Raman cross-sections of the 2 different molecules. These challenges were circumvented by using a molecule and its isotope-edited analog as the bianalyte pair that have the same affinities and Raman cross-sections.⁴⁻⁶ Therefore, the BiASERS distributions of both different nucleobases and the cytosine isotopologues provide strong evidences to single-molecule detection of all 4 nucleobases by our platform.

Figure R1. BiASERS analysis of SERS time series of trapped $C_{15}C$ -AuNUs. (a) SERS spectra of trapped multilayer C -AuNU, multilayer $C_{15}C$ -AuNU, single-molecule C -AuNU and single-molecule $C_{15}C$ -AuNU, respectively. The spectra intensities were adjusted to highlight the peak positions. (b) SERS time series of submonolayer $C_{15}C$ -AuNUs trapped in gold nanoholes. (c) Representative single-molecule spectra of the C -AuNU, $C_{15}C$ -AuNU and many-molecule spectrum of $C_{15}C$ -AuNU, respectively. (d) BiASERS distribution of $C_{15}C$ and C out of about 1100 spectra.

Corresponding revision in Page 12- 14 of the revised manuscript:

“We firstly used the bi-analyte SERS technique (BiASERS) with Cytosine (C) and isotope-edited Cytosine ($C_{15}C$) molecules to demonstrate single-molecule capability of our platform.^{5, 6} In comparison to C , the $C_{15}C$ has some carbon and nitrogen atoms replaced by their isotopes (Supplementary Table S2). Importantly, both C and $C_{15}C$ exhibit the same surface area (1.27 nm^2), Raman cross-section and affinity to

gold surface. Due to the isotope atoms, the SERS peak of C_{iso} at around 941 cm^{-1} was shifted from that of C at around 904 cm^{-1} (Supplementary Figure S8). To demonstrate single-molecule SERS, sub-monolayer of equal moles of C and C_{iso} molecules were adsorbed on the AuNU (C_{iso} C-AuNU) with a surface coverage around 1 molecule per 4 nm^2 . As shown in Figure 4b, one SERS peak appearing at the C range ($900 - 930\text{ cm}^{-1}$) represented detection of single C molecule, and the emergence of one SERS peak at the C_{iso} range ($930 - 960\text{ cm}^{-1}$) represented detection event of single C_{iso} molecule. If both peak appeared, it belonged to multi-molecule events.

The BiASERS analysis required >1000 events (SERS spectra) to statistically prove single-molecule SERS (see Methods for data processing details), which actually depended on a sensitive and localized hot spot on the AuNU tip. If the hot spot has a small size to cover only 1 molecule, the probabilities of detecting either one of the molecules are high. If the hot spot became large enough to cover >1 molecules, the probability of detecting both of them increases as well. When the former are larger than the latter in our case of around 1030 events, a BiASERS histogram with more single-molecule events than multi-molecule events (Figure 4c) was established to confirm single-molecule detection.

Figure 4. BiASERS analysis of single-molecule SERS of individual nucleobases. (a) SERS time series of submonolayer C_{iso}C-AuNUs trapped in gold nanoholes. (b) Representative single-molecule spectra and many-molecule spectrum of C_{iso}C-AuNU, respectively. (c) BiASERS distribution of C_{iso} and C out of about 1030 spectra. (d) SERS time series of the submonolayer AG-AuNUs trapped in a nanohole. (e) Typical SERS spectra of single-molecule events and many-molecule events of AG-AuNU. (f) BiASERS distribution of A and G out of about 1200 spectra. (g) SERS time series of the CT-AuNUs trapped in a nanohole. (h) Typical SERS spectra of single-molecule events and many-molecule events of CT-AuNU. (i) BiASERS distribution of C and T from about 1500 collected spectra. All color bars represent the signal-to-baseline intensity of the SERS peaks. The dotted lines indicate SERS peak positions of multilayer nucleobases (black for A or G, red for C or T) adsorbed on silver nanoparticles without bias, respectively, from Ref.⁷

Also they use the argument that the orientation is resulting in shifted Raman peak positions, to plead for single molecule characteristics. It has been shown before in single particle SERS (also in plasmonic nanoholes - Kerman et al, *Nanoscale*, 2015,7, 18612), that this type of effects occur, and are likely related to temporal adsorption to the gold surface, and the resulting stress that leads to the peak shifts (this can be the case too for monolayers).

Reply: We thank the referee for his/her concern and suggestion. The SERS peak shifts of single molecules is a mixed effect due to many mechanisms, such as changes of molecule orientations on the metal surface⁸⁻¹⁰ and molecule diffusion on the nanoparticle surface^{11, 12}. The effect in our case of measuring single molecules in the electroplasmonic and fluidic environment is even more complicated. The suggested paper [Kerman et al, *Nanoscale*, 2015,7, 18612] used a plasmonic nanohole of rectangle shape to trap, and measure SERS signals of, a single polystyrene nanoparticle that diffused through the rectangle nanohole.¹³ Although the molecules measured were different from our case of submonolayer nucleotide molecules on the AuNU, a trapped AuNU near the wall of the gold nanohole could be pulled by similar plasmonic force to touch the nanohole wall temporarily. We cited the paper and added more discussions.

In the case that the AuNU was pulled closed to the nanohole wall, the electromagnetic field at the AuNU tip would become more localized and stronger with decreasing distance between the AuNU and the nanohole sidewall (AuNU-sidewall distance), as shown in Figure R2 below. The AuNU tip mimics a gold nanotip moving close to a gold film in tip-enhanced Raman spectroscopy (TERS) experiments that could induce accordingly following effects to change the SERS spectra. 1) The effect of local field gradient is possible that the local electromagnetic field can change intensity greatly over a few angstrom.¹⁴ The field gradient effect could induce electric dipole–electric quadrupole polarizability tensor of the adsorbed molecules. As a result, the Raman selection rule changed and Raman-inactive bands became active.^{9, 15, 16} 2) The increased plasmonic force at the AuNU tip due to the increased field gradient could pull the adsorbed molecule and change its orientation.¹⁷ 3) The local temperature at the AuNU tip could increase to allow fast molecule motions,¹⁸ including rotation and diffusion on the AuNU tip.^{10, 19} 4) The adsorbed molecule can be deformed by the interaction between the electric double layers on the AuNU tip and the alumina layer on the nanohole wall, respectively. Peak shift up to 15 cm⁻¹ of the ring-breathing mode of adenine molecules pressed by a gold nanotip was reported.^{20, 21} 5) The oscillation of the trapped AuNU in the nanohole will change the AuNU-sidewall distance as well as induce lateral drifting. If the adsorbed molecule is pressed by

the electric double layers, the drifting may change its orientation on the AuNU tip and lead to the SERS peak shifts.

Figure R2. Simulated field intensity enhancement and distribution on the AuNU tip with different gap distance (G) between the AuNU tip and the alumina layer coated on the nanohole, suggesting that smaller gap leads to stronger and more confined electromagnetic fields.

Corresponding revision of the revised manuscript:

Page 10: “The trapping time of single AuNU was actually affected by an interplay between the AuNU hot spot and the trapping force. The AuNUs were not uniform in sizes and shapes, leading to slight difference in the distance between the AuNU tip and nanohole wall. This in return influenced the hot spot strength, because smaller distance gave rise to a stronger hot spot and vice versa. When the AuNU was pulled close to the sidewall, the coupling between the nanohole and the AuNU tip became stronger to enhance further the optical force. As a result of this positive cycle of the optical force, some AuNUs may be pulled to even touch the nanohole wall temporarily, which could induce strong interactions to affect the SERS spectra as well as the trapping time.^{13, 32}”

Page 16: “When the AuNU-sidewall distance is short enough, the AuNU tip mimics the gold nanotip moving close to a gold film in TERS experiments. Accordingly, following effects could be induced to change the SERS spectra. 1) The effect of local field gradient is possible that the local electromagnetic field can change intensity greatly over a few angstrom.¹⁴ The field gradient effect could even induce electric dipole–

electric quadrupole polarizability tensor of the adsorbed molecules to activate Raman-inactive bands.^{15, 16} 2) The increased plasmonic force at the AuNU tip due to the increased field gradient could pull the adsorbed molecule.¹⁷ 3) The local temperature at the AuNU tip could increase to allow fast molecule motions,¹⁸ including rotation and diffusion on the AuNU tip.^{10, 19} 4) The adsorbed molecule can be deformed by the interaction between the electric double layers on the AuNU tip and the alumina layer on the nanohole wall, respectively. 5) The oscillation of the trapped AuNU in the nanohole may induce lateral drifting of the AuNU tip. If the adsorbed molecule is pressed by the electric double layers, the drifting may change its orientation on the AuNU tip.”

Overall: the results show interesting indications, but not real proof yet for single molecule sensitivity. In addition, they could indeed be used for SERS detection of small molecules, but the general applicability for a broader set of markers is questionable.

Reply: As reported above, thanks to the new BiASERS experiments, we think that we provided a clear demonstration of single molecule detection. In fact, this claim is now supported by the BiASERS distributions of both different nucleobases and the cytosine isotopologues as suggested by the reviewer. It is not clear what the range of “markers” here is, but we understand the concern that the proteins seem too large to be covered by the hot spots on the AuNU tips. Small molecules such as amino acids with sizes and Raman cross-sections similar to the nucleobases can be discriminated by our platform.^{22, 23} On the other hand, structural and conformational information of proteins has been obtained by analyzing SERS spectra from amino acids in subdomains of the protein molecules attached on silver or gold nanoparticles.²⁴⁻²⁶ In fact, our platform can trap either AuNUs or gold nanospheres to obtain hot spots of different sizes to cover one or more amino acids of a protein molecule. Therefore, we think that our method of trapping and detecting a protein-adsorbed gold nanoparticle in a PBS solution will be promising to monitor the dynamic process of proteins in the aqueous environment. We added this discussion to the manuscript to clarify the potential applications of our method.

Corresponding revision in Page 22 in the revised manuscript:

“Meanwhile, structural and conformational information of proteins has been obtained by analyzing SERS spectra from amino acids in protein subdomains.²⁴⁻²⁶ By trapping gold nanospheres, our method can also

obtain hot spots of different sizes to cover one or more amino acid residues of a protein molecule. Therefore, our method can also be a promising SERS platform to detect amino acids and proteins.”

Reviewer #2 (Remarks to the Author):

The authors propose a plasmonic nanohole assisted trapping system to detect SERS signal of DNA bases at a single molecular level. Gold nano-stars could be trapped by the combined effect of electroosmotic, electrophoretic, and optical forces. Due to hot spots excited in the gap between the AuNU and hole sidewall, it is able to distinguish different molecules with reliable and stable results. However, the Raman signals in this work seem stochastic rather than controllable when multiple samples are mixed, and there have been many approaches to get a simple single molecular Raman spectrum in the plasmonic tweezers.

Reply: Thank you for your comments. We have prepared point-to-point replies below and included them in our revised manuscript for your reconsideration. In particular, we notice that the “stochastic” origin of the presented data relies on the single-molecule nature of the experiments. Also, we notice that indeed we observed sub-molecular features within a single molecule. In this kind of experiments, signal fluctuations are unavoidable and usually considered as a proof of single-molecule nature.⁹ In our opinion, the key point is not to prevent signal fluctuations (that are very hard to be avoided due to the nature of experiment) but rather to achieve reliable and stable data. We thank the Reviewer for having noticed that we reached the latter point.

My major concerns:

1. It has widely been demonstrated that hot spots could also be excited at the tip of Au nanourchins even without the nanoholes (The Journal of Physical Chemistry C, 2008, 112(48): 18849-18859; Applied Physics Letters, 2009, 94(15): 153113; J. Am. Chem. Soc. 2015, 137, 33, 10460-10463; etc.). As a result, the authors should provide comparable results without the plasmonic nanohole, to highlight its key role of the hole plays in this system.

Reply: We agree with Reviewer that there are many ways to achieve single-molecule SERS spectra and the use of nanourchins is one of that. For completeness, we added the suggested papers to the references. However, we notice that the suggested papers mainly report single-particle SERS spectra rather than single-molecule SERS spectra. Furthermore, our flow-through systems enable the analyses of complex oligonucleotides with ultimate detection limits, which are generally lower than that achieved with nanourchins dispersed in solution. Also, our manuscript reports sub-molecular features that are very hard to be achieved by using only nanourchins, even though it is in principle possible. For comparison, we followed the suggestion of the Reviewer to carry out experiments without the use of nano-holes, as shown in Figure R3 below. Figure R3a shows SERS time series of 9ACTG-AuNUs diffusion in solution, which are not

reproducible. Although the SERS peaks can be assigned to nucleotides (Figure R3b), the changing peaks position and intensity can neither recognize them from one single nucleotide of the 9ACTG molecule nor be used for further analysis such as the molecule conformation.

Figure R3. (a) SERS time series of 9ACTG-AuNUs diffusion in solution; the color bar represents the peak intensity. (b) SERS spectra extracted from (a); the arrows indicate the times when the spectra were collected; the dashed lines indicate the peak positions.

Corresponding revision in Page 3 in the manuscript:

“The AuNUs had many sharp tips that could confine electromagnetic fields for SERS detections.”²⁷⁻²⁹

2. In Fig.4, the Adenine and Guanine are mixed with equal moles (also Cytosine and Thymine), why the measured number of G/C is almost twice of A/T? And, the concentration of C-T is much lower than A-G, why the detected C-T's peak intensity is much higher?

Reply: In common BiASERS analysis of >1000 SERS spectra of silver colloids, colloid clusters were instantly formed during their diffusion in solutions.² Such >1000 SERS spectra came probably from similar amount of different colloid clusters, and the peak distribution in the BiASERS histogram should agree with the different nucleobase affinities to gold surface (A > C ≥ G > T). In strong contrast, the >1000 SERS spectra for the BiASERS analysis in our method came from a few single AuNUs trapped in our platform for

tens of seconds. As a result, the fact that the measured number of G/C is almost twice of A/T can be ascribed to the different nucleobase affinities to gold surface as well as different trapping time of the AuNUs. As shown in Figure R4a below, AG-AuNU 1 and AG-AuNU 2 exhibited only 1 peak at the G range (660-700 cm^{-1}). The time that the Single-G peak appeared corresponds to the trapping time of the AG-AuNU. Hence, the trapping time of AG-AuNU 1 and AG-AuNU 2 were longer than AG-AuNU 3 and AG-AuNU 4 that exhibited only 1 peak at the A range (700-740 cm^{-1}). This led to more single-G events than single-A events in the BiASERS histogram (Figure R4c). Similarly, the CT-AuNU with single-C peak was trapped in 3.1 – 7.5 min, which was longer than the CT-AuNU with Single-T peak during 2 – 2.6 min in the case of CT-AuNUs (Figure R4d). As a result, more single-C events than single-T events were found in the BiASERS histogram (Figure R4f).

The higher C-T's peak intensity in the "Both" spectrum in Figure R4e is probably due to the multi-molecule events of many C and T molecules excited in a hot spot on the AuNU tip. We can approximate the maximum hot spot size as $\leq 2 \text{ nm}^2$ because the BiASERS histogram of AG-AuNUs (Figure R4c) was achieved with a surface coverage of 1 A or G molecule per 2 nm^2 . The surface areas of the C (1.27 nm^2) and T (1.42 nm^2) are equal to or smaller than those of A (1.42 nm^2) and G (1.54 nm^2). Therefore, more C molecules could be covered by such hot spot. Furthermore, C adsorbed on gold surface with an angle tilted from the surface in solution phase,³⁰ further increasing the number of C in the hot spot. To achieve the BiASERS histogram (Figure R4f), the number of the multi-molecule events was reduced by decreasing the surface density of C and T on the AuNU to 1 molecule per 7 nm^2 .

Figure R4. BiASERS analysis of SERS time series of individual nucleobases. (a) SERS time series of the submonolayer AG-AuNUs trapped in a nanohole. (b) Typical SERS spectra of single-molecule events and mixed events of AG-AuNU. (c) BiASERS distribution of A and G out of about 1200 spectra. (d) SERS time series of the CT-AuNUs trapped in a nanohole. (e) Typical SERS spectra of single-molecule events and mixed events of CT-AuNU. (f) BiASERS distribution of C and T from about 1500 collected spectra. All color bars represent the signal-to-baseline intensity of the SERS peaks. The dotted lines indicate SERS peak positions of multilayer nucleobases (black for A or G, red for C or T) adsorbed on silver nanoparticles without bias, respectively, from Ref.⁷

Corresponding revision in the manuscript:

Page 15: “...The fact that more single-G events than single-A events was due to different nucleobase affinities to gold surface ($A > C \geq G > T$)¹ as well as different trapping time of the AuNUs. For example, the AG-AuNU 2 exhibiting only 1 peak at the G range (660-700 cm⁻¹) was trapped longer than the AG-AuNU 3 that exhibited only 1 peak at the A range (700-740 cm⁻¹).”

Page 17-19: “...Yet, the surface areas of the C (1.27 nm²) and T (1.42 nm²) are smaller than those of A and G. Furthermore, C adsorbed on gold surface with an angle tilted from the surface in solution phase.³⁰ As a result, many C and T molecules were covered in a hot spot on the AuNU tip that produced higher peak intensity, such as the “Both” spectrum in Figure 4h. Therefore, the BiASERS distribution of the CT-AuNUs

(Figure 4i) was not achieved until the CT coverage on the AuNU surface was decreased to 1 molecule per 7 nm^2 in order to lower the probabilities of including >1 molecules in the hot spot. Long trapping was observed as the Cytosine peak intensity at around 685 cm^{-1} fluctuated from 3.1 to 7.5 min under a 1V bias and a 6 mW laser power, leading to more single-C events than single-T ones.

... Besides, in common BiASERS detection of >1000 SERS spectra by, for example, silver colloids, colloid clusters were instantly formed during their diffusion in solutions.² Such >1000 SERS spectra came probably from similar amount of different colloid clusters that had different field enhancement, hot spot size and molecule adsorption in the hot spots. In strong contrast, the >1000 SERS spectra for the BiASERS analysis in our method came from a few single AuNUs trapped in our platform for tens of seconds respectively. The resultant hot spots were formed after the molecule adsorption to ensure that the same molecule were excited reproducibly by the same intense hot spot with similar size.”

3. In Fig. 4a, there exhibits considerable peak shifts for AG-AuNUs, it makes sense to attribute to molecule reorientation on the AuNU surface. But, how to explain the two peaks shifting in opposite directions?

Reply: The shifting of the two peaks to opposite directions might be due to the fact that the A/G molecules were in different initial states of orientation on the AuNU tip. For example, in a recent report of cryogenic tip-enhanced Raman spectroscopy of single molecule dynamics at 90 K (Figure R5 below),¹⁰ the malachite green molecule was observed to have 3 different orientation states on a gold film (Figure R5d), which corresponds to different positions of the Raman peaks at 525 and 799 cm^{-1} . These peaks blue shifted when the malachite green molecule was initially at the orientation state of 46 s and rotated to the state of 48 s. They red shifted when the malachite green changed from the state of 48 s to the state of 50 s, i.e. back to the same state of 46 s. Such molecular reorientations were induced by the ambient thermal energy and could happen faster in room temperature.¹⁰ Similarly, the A/G molecules on the AuNU could have different initial orientation states in the more energetic environment of the electroplasmic trap.

Figure R5. (a) Schematic of the cryogenic TERS experiment on single malachite green molecule on a gold film. (b) Contour map of 90K TERS spectra of the malachite green molecule. (c) Subset spectra (dots) from the dashed box in (b) with Lorentzian fits (solid) exhibit shifting of peaks at 525 and 799 cm^{-1} (B_1 mode) versus 528 and 804 cm^{-1} (A_2 mode). (d) Corresponding molecular orientations at each time slice simulated by DFT.

Corresponding revision in Page 17 in the manuscript:

“Furthermore, the peaks of the AG-AuNU 2 and AG-AuNU 3 that shifted to opposite directions could be ascribed to different initial states of orientations on the AuNU tip. For example, rotations of non-symmetrical molecules on a gold film at 90K were observed to shift Raman peaks in opposite directions due to reversible orientation states. Such molecular reorientations were induced by the ambient thermal energy and could happen faster at room temperature.¹⁰ Similarly, the A and G molecules on the AuNU at different initial orientation states in the more energetic environment of the electroplasmic trap could give rise to the opposite peak shifting.”

4. The author mentioned that there will be ‘a strong fluctuation of both the peak positions and the baseline’ when there are multiple particles trapped, I wondered if the fluctuations of the peak and baseline are synchronous, as I find that there is an obvious mismatch at the time of ~670s and ~850s in Figure S5.

Reply: When 1 AuNU was trapped, the peak at 730 cm^{-1} and baseline at 1000 cm^{-1} are synchronous due to the stable spectra from the trapped AuNU. They had mismatch due to changes of peak intensities and positions (Figure R6 below). The measurements of the baseline at 1000 cm^{-1} and the peak height at 730 cm^{-1} of a SERS spectrum of Figure S5 are shown in Figure R6b below. In comparison to the spectrum at 669.0 s, the baseline at 1000 cm^{-1} suddenly increases while the 730 cm^{-1} peak height remains similar in the spectrum at 672.5 s (Figure R6c), leading to the time trace mismatch during 670 to 680 s.

Figure R6. (a) Magnified view of time traces of SERS signals of the trapped multilayer A-AuNUs during 655 to 695 s in Figure S5a; At 660s, 1V bias was applied to trapped the A-AuNUs. The SERS spectra of trapped AuNUs at (b) 669.0 s and 672.5 s(c).

Mismatches as such were used to characterize the existence of multi AuNUs. When the 1V bias was turned on at 660 s, multi AuNUs were driven and crowded in the nanohole. As a result, many strong gap-based hot spots due to either inter-AuNU coupling or coupling between AuNU tips and the nanohole wall were generated. However, the crowded AuNUs in the nanohole were in a dynamic movement for a short time due to the Brownian motion. Weak modes could be enhanced randomly by those hot spots to change the peak intensities and positions of the spectra, such as the one in Figure R6c. Later, when the AuNUs left the nanohole and only 1 AuNU remained, the system reached a stable trapping of 1 AuNU. Consequently, only one gap-based hot spot of the trapped AuNU tip near the nanohole sidewall remained and became dominant, so the adenine on this tip surface were continuously excited to produce reproducible and stable spectra, resuming synchronous time traces. Similarly, the time trace mismatch at around 848 s in Figure S5 can be ascribed to that fact that multi AuNUs translocated through a nanohole that already trapped a single AuNU in it. As shown in Figure R7 below, the spectrum at 848.3 s with changed baseline and peak positions suggested formation of many gap-based hot spots due to the short time that multi AuNUs moved in and out of the nanohole. Afterwards, the spectrum resumed reproducible and the time trace resumed synchronous from 848.9 s.

Figure R7. (a) Magnified view of time traces of SERS signals of the trapped multilayer A-AuNUs during 815 to 875 s in Figure S5a; During 825 and 860 s, 1V bias was applied to trap the AuNU. (b) The SERS spectra of trapped AuNUs at 846.8, 848.3 and 848.9 s.

Corresponding revision in Page 9 in the revised manuscript:

“If more than one A-AuNUs were driven in the nanohole, many strong gap-based hot spots were generated due to either inter-AuNU coupling or coupling between AuNU tips and the nanohole wall. However, the conformations of the adenine molecules adsorbed in these hot spots were not necessarily the same. Weak modes could be enhanced randomly by those hot spots to change the peak intensities and positions of the spectra, leading to strong fluctuation of both the peak positions and the baseline (Supplementary Figure S6).³¹ When only 1 AuNU remained stably trapped in the nanohole, only one gap-based hot spot of the trapped AuNU tip near the nanohole sidewall remained and became dominant. Thus, the adenine on this tip surface were continuously excited to produce reproducible and stable spectra.”

5. For the sample 9C1A in Fig.5a, I suppose the Raman peak of Cytosine will be much easier to be detected as there are more C than A. But I find a reversed result. The authors gave out an explanation that the affinity of A was the biggest in the four bases. If this is the right reason, for the sample of 1C9A, I think, it will be more difficult to detect the signal of Cytosine. However, in Figure S8, I find that the Cytosine’s signal seems even easier to be detected compared with Adenine. For this, I think the discussion is not strong enough to support such a conclusion.

Reply: We thank the referee for his/her concerns. In fact, we notice that among so many SERS spectra of the oligonucleotides, only small part of them demonstrated single nucleobases: about 6.25% single A data

in the 9C1A spectra and only 0.16% single C data in the 1C9A spectra, respectively. In the case of 9ACTG, there are 1.41% single C, 2.10% single T and 4.75% single G data, respectively. The detection probabilities of each nucleobases agrees partially with the nucleobase affinities to gold surface ($A > C \geq G > T$)¹, and the rest depends on the different trapping time of the AuNUs as shown in Figure R8 below. In comparison to the nucleobase-AuNUs, the Zeta potentials of the oligonucleotide-adsorbed AuNUs (Table S1 below) were not uniform, suggesting inhomogeneous adsorption presumably due to the negatively charged phosphate backbond.¹ As a result, the optical force dominated the trapping of the oligonucleotide-adsorbed AuNUs. However, the AuNUs were not uniform in sizes and shapes, leading to slight difference in the distance between the AuNU tip and nanohole wall. This in return influenced the hot spot size because smaller distance gave rise to a stronger and more localized hot spot and vice versa. When the AuNU was pulled close to the sidewall, the coupling between the nanohole and the AuNU enhanced became stronger to enhance further the gradient force, which may pull the AuNU further to touch the nanohole wall temporarily.^{13, 32} The hot spot size and the trapping force thus interplayed to affect the trapping time.

Figure R8. SERS time series 1C9A-AuNUs (a) and 9ACTG-AuNUs (b - d) to demonstrate sensing of (a, b) single Cytosine, (c) single Guanine, and (d) single Thymine, respectively. The color bars represent the peak intensity. The arrows indicate the frequency positions of the Raman peaks assigned to either A (black), C (red), T (green) or G (blue).^{7, 33} Single-base events are indicated by the long color arrows.

Table S1. Measured Zeta potentials (ζ_{np}) of non-functionalized, multilayer and sub-monolayer nucleobase-AuNUs solutions.

		ζ_{np} 1(mV)	ζ_{np} 2(mV)	ζ_{np} 3(mV)
Non-functionalized	AuNU	-21	-23	-23.4
Multilayer	A-AuNU	-17.8	-20.4	-19.6
	C-AuNU	-19.4	-22.5	-23.2
	C _{iso} -AuNU	-17.2	-17.8	-16.7
	T-AuNU	-18.2	-22.1	-23.4
	G-AuNU	-19.9	-19.7	-20.2
Submonolayer	AG-AuNU	-16.5	-16.1	-14.6
	CT-AuNU	-24.0	-21.1	-16.9
	C _{iso} C-AuNU	-22.8	-21.0	-21.8
	1C9A-AuNU	-25.8	2.48	-52.9
	9C1A-AuNU	-33.4	-19.2	-51.3
	9ACTG-AuNU	1.18	-18.8	-32.7

Corresponding revision in the manuscript:

Page 10: “The trapping time of single AuNU was actually affected by an interplay between the AuNU hot spot and the trapping force. The AuNUs were not uniform in sizes and shapes, leading to slight difference in the distance between the AuNU tip and nanohole wall. This in return influenced the hot spot strength, because smaller distance gave rise to a stronger hot spot and vice versa. When the AuNU was pulled close to the sidewall, the coupling between the nanohole and the AuNU tip became stronger to enhance further the optical force. As a result of this positive cycle of the optical force, some AuNUs may be pulled to even touch the nanohole wall temporarily, which could induce strong interactions to affect the SERS spectra as well as the trapping time.^{13, 32}”

Page 21: “Among so many SERS spectra of the oligonucleotides, only small part of them demonstrated single nucleobases: about 6.25% single-A data in the 9C1A spectra and only 0.16% single-C data in the 1C9A spectra, respectively. In the case of 9ACTG, there are 1.41% single-C, 2.10% single-T and 4.75% single-G data, respectively. The detection probabilities of each nucleobases agrees partially with the nucleobase affinities to gold surface ($A > C \geq G > T$)¹, and the rest depends mainly on the different trapping time of the AuNUs.”

Given this, I cannot recommend its publication in Nature Communications.

Reviewer #3 (Remarks to the Author):

The manuscript by Huang et al demonstrated single molecule SERS detection of DNA bases in oligonucleotides. The detection method was based on trapping single gold nanourchin (AuNU) into a plasmonic nanohole to form a greatly enhanced electromagnetic field, which facilitates the detection of biomolecules of small Raman Scattering cross sections. The confinement of the AuNU inside the nanohole was achieved through long-time electro-plasmonic trapping, which is desirable for flow-through SERS detection. While the manuscript is interesting and the concept advances the field, the authors should consider performing a few more experiments to make a compelling case.

Reply: We thank the referee for the comments. We have prepared point-to-point replies below and included them in our revised manuscript for your reconsideration.

1. In Figure 3b, the author showed the contour map of the Raman spectra produced by a trapped A-AuNU. The on and off presence of the Raman signal is achieved through turning on and off the bias. However, a significant increase in the peak intensity is noticed consistently for the third and fourth ON periods. Is it possible that more AuNUs are getting trapped during those periods?

Reply: We believe that the increase of peak intensity for the 3rd and 4th ON periods at Figure 3b were not due to trapping more AuNUs in the nanohole. Rather, it could be probably due to trapping of a single AuNU with shorter gap distance between the AuNU tip and the nanohole wall than the one in the 2nd ON period. In fact, the simulation of the electromagnetic field distribution on the tip surface in Figure R2 below shows that shorter gap distance can provide larger field enhancement and thus stronger SERS signals. Hence, a stronger electromagnetic coupling between the nanohole wall and the AuNU can lead to variations of peak intensities.

Figure R2. Simulated field intensity enhancement and distribution on the AuNU tip with different gap distance (G) between the AuNU tip and the alumina layer coated on the nanohole, suggesting that smaller gap leads to stronger and more confined electromagnetic fields.

The fact that the SERS spectra were stable during the 3rd and 4th ON periods removes the possibility of trapping more than 1 AuNU in the nanohole. In the case of more than 1 AuNU were driven to the nanohole when the bias was turned on, the AuNUs would crowd into the nanohole to generate many strong hot spots due to either inter-AuNU coupling or coupling between AuNU tips and the nanohole wall. In such a condition, the SERS spectra of > 1 AuNU crowding in the nanohole exhibited violent changes of peak positions and intensities, such as those between 660 to 690 s in Figure R9 below. When only 1 AuNU remained stably trapped in the nanohole, only one gap-based hot spot of the trapped AuNU tip near the nanohole sidewall remained and became dominant. The molecule on this tip surface were continuously excited to produce reproducible and stable spectra, such as those between 730 and 780 s in Figure R9 below.

Figure R9. (a) Time trace of A-AuNU trapped and released in a nanopore by 1V bias and laser power 12 mW. (b) The contour map of the corresponding Raman spectra produced by the trapped A-AuNU. At 660s, trapped AuNUs produced irreproducible SERS signals as well as fluctuating baseline until 690s, which may be because there is >1 AuNUs in the nanopore. After 690s, the baseline became stable, suggesting only one AuNU was left trapped in the nanopore.

Corresponding revision in Page 9 in the manuscript:

“We notice that, among 4 trapping periods, the 3^d trapping period between 100 - 170 s exhibited higher peak intensity than those of the previous 2 periods. It could be due to trapping of a single AuNU with shorter gap distance between the AuNU tip and the nanopore wall, which provide larger field enhancement (Supplementary Figure S3). Furthermore, another reason could be stronger electromagnetic coupling between the nanopore wall and another sharper tip of the same AuNU.

The stable and reproducible SERS spectra (Figure 3b) suggested the trapping of a single A-AuNU rather than more than one A-AuNU. If more than one A-AuNUs were driven in the nanohole, many strong gap-based hot spots were generated due to either inter-AuNU coupling or coupling between AuNU tips and the nanohole wall. However, the conformations of the adenine molecules adsorbed in these hot spots were not necessarily the same. Weak modes could be enhanced randomly by those hot spots to change the peak intensities and positions of the spectra, leading to strong fluctuation of both the peak positions and the baseline (Supplementary Figure S6).³¹ When only 1 AuNU remained stably trapped in the nanohole, only one gap-based hot spot of the trapped AuNU tip near the nanohole sidewall remained and became dominant. Thus, the adenine on this tip surface were continuously excited to produce reproducible and stable spectra.”

2. In the caption of Figure 1f, the authors indicate that it is magnetic field distribution. Why did the authors choose to the magnetic field instead of electric field? If it is magnetic field, the authors have to include a color scale bar and discuss the significance of the magnetic field distribution in this context.

Reply: I am afraid that it is a misunderstanding. The caption of Figure 1f was written “Magnified field distribution...”. To make it more clear, we changed it as “Magnified view of the intensity distribution of the electromagnetic field at one tip of the AuNU.” in Page 4 in the revised manuscript.

3. It would be better to include more information related to the zeta potential of the AuNU before and after modifying various analytes. Since zeta potential plays a critical role for colloidal stability and electro-plasmonic trapping, it might be useful for the reader know the values.

Reply: We thank the referee for the advice. The Zeta potentials of the non-functionalized AuNU solutions below are added to Table S1 of the supporting information.

Table S1. Measured Zeta potentials (ζ_{np}) of non-functionalized, multilayer and sub-monolayer nucleobase-AuNUs solutions.

		ζ_{np} 1(mV)	ζ_{np} 2(mV)	ζ_{np} 3(mV)
Non-functionalized	AuNU	-21	-23	-23.4
Multilayer	A-AuNU	-17.8	-20.4	-19.6
	C-AuNU	-19.4	-22.5	-23.2
	C _{iso} -AuNU	-17.2	-17.8	-16.7
	T-AuNU	-18.2	-22.1	-23.4

	G-AuNU	-19.9	-19.7	-20.2
Submonolayer	AG-AuNU	-16.5	-16.1	-14.6
	CT-AuNU	-24.0	-21.1	-16.9
	C _{iso} C-AuNU	-22.8	-21.0	-21.8
	1C9A-AuNU	-25.8	2.48	-52.9
	9C1A-AuNU	-33.4	-19.2	-51.3
	9ACTG-AuNU	1.18	-18.8	-32.7

4. The author has mentioned that all the analytes are absorbed on the AuNU through physical absorption. Hence, more experimental evidence of the efficient co-absorption and co-presence of analytes (such as Adenine and Guanine) on the AuNU is needed to further support the SERS spectra obtained from single-molecule measurement. The electromagnetic field intensity achieved by trapping the AuNU is within the range that can be achieved normal electromagnetic hotspots formed between assemblies of nanoparticles. Maybe the authors can harness such hotspots to further understand the adsorbate composition.

Reply: We thank the referee for this concern. To be clear, by physical adsorption, we meant non-covalent self-assembly of the analyte directly on the AuNU. Such direct adsorption of the nucleotide on gold nanoparticles was well studied in literature, and in our opinion should be considered well established. It does include chemical bonding, hydrophobic force and electrostatic force.^{1,34} To be more precise, we have replaced “*physical adsorption*” by “*direct adsorption*” everywhere in the revised manuscript.

We did estimate successful co-adsorption of different nucleotides on the AuNUs by comparing the Zeta potentials with that of non-functionalized AuNUs (Table S1) before proceeding to the SERS experiments. The non-functionalized AuNUs were stabilized by negatively charged citrates that were weakly adsorbed on the AuNUs and exhibited a Zeta potential around -22.5 mV.³⁴ Also, the molecular structures and SERS spectra of both the citrate and the nucleobases are well-known: no citrate peaks are overlapped with those of the nucleobases in the DNA fingerprint range (500 – 1600 cm⁻¹).^{7,35} Our detection of a base was confirmed by successfully assigning at least 2 Raman modes, which unambiguously proved the presence of the base.

5. We suggest SERS measurement of AuNU alone as a negative control.

Reply: The Raman spectra of the non-functionalized AuNUs that diffused in solution are shown as the colored curves in Figure R10c below. They may originate from the citrates that were loosely adsorbed on the AuNUs for stabilization. The difference of the spectra baselines were due to the non-uniform shapes of the AuNUs (Figure R10a,b). In the data processing of the single-molecule SERS spectra, the baselines were fitted by 5th-order polynomial functions and then subtracted from the spectra.

Figure R10. (a,b) TEM images of the AuNUs. (c) Measured Raman spectra of the non-functionalized AuNUs diffusion in solution, in which the black curve was the background of the solution.

Corresponding revision in the manuscript:

Figure R10 was added to the Supporting information as Figure S2.

6. Also, based on all the experiments performed, which serve as training set, it would be great to design and implement a few double-blinded studies to demonstrate that the SERS spectra can provide accurate determination of the DNA bases in unknown oligos. This will make a compelling case that the demonstrated approach is indeed powerful for the detection of single bases in DNA.

Reply: We appreciate the suggestion for improving our manuscript. However, an accurate determination of the DNA bases is related to quantitative SERS experiment to determine the number of the DNA bases in an oligonucleotide, which is beyond the scope of our current manuscript. Due to the foreseeable large amount of data and analysis for the double-blinded studies, we will include them in another manuscript devoted to practical implementations.

On the other hand, to strengthen our manuscript, we performed additional single-molecule experiment with cytosine (C) and isotope-edited cytosine (C_{iso}) molecules submonolayer adsorbed on the gold nanourchins (C_{iso}C-AuNUs) for BiASERS analysis. This kind of experiment is considered an unambiguous proof of single molecule.³⁶ Both C and C_{iso} have the smallest area (1.27 nm²) among the 4 nucleobases and 6181 Cytosines are needed to form a monolayer on a AuNU of 50 nm in diameter. Therefore, we designed

the experiment by allowing adsorption of 2000 molecules at maximum ($1000 C_{\text{iso}} + 1000 C$) on one AuNU, which was approximately 1 molecule per 4 nm^2 . The SERS mode of the multilayer C_{iso} -AuNU at around 950 cm^{-1} is shifted from that of multilayer C-AuNU at around 904 cm^{-1} due to the isotope atoms, as shown in Figure R1a below. This mode was chosen for our BiASERS analysis because it has high intensity at the concentration level of submonolayer without neighbouring peaks to affect fitting its position. In the SERS spectra of the trapped C_{iso} C-AuNUs, the numbers of peaks at either the C range ($900 - 930 \text{ cm}^{-1}$ representing only C) or the C_{iso} range ($930 - 960 \text{ cm}^{-1}$ representing only C_{iso}) are much larger than the number of 2 peaks appearing at both ranges. Such BiASERS distribution (in Figure R1d below) proved single-molecule detection by our particle trapping method.

Figure R1. BiASERS analysis of SERS time series of trapped C_{iso} C-AuNUs. (a) SERS spectra of trapped multilayer C-AuNU, multilayer C_{iso} -AuNU, single-molecule C-AuNU and single-molecule C_{iso} -AuNU, respectively. The spectra intensities were adjusted to highlight the peak positions. (b) SERS time series of submonolayer C_{iso} C-AuNUs trapped in gold nanoholes. (c) Representative single-molecule spectra of the C-AuNU, C_{iso} -AuNU and Both. (d) Bar chart showing the number of peaks in the C, Both, and C_{iso} ranges.

many-molecule spectrum of C_{iso} C-AuNU, respectively. (d) BiASERS distribution of C_{iso} and C out of about 1100 spectra.

Corresponding revision in Page 12- 14 of the revised manuscript:

“We firstly used the bi-analyte SERS technique (BiASERS) with Cytosine (C) and isotope-edited Cytosine (C_{iso}) molecules to demonstrate single-molecule capability of our platform.^{5, 6} In comparison to C, the C_{iso} has some carbon and nitrogen atoms replaced by their isotopes (Supplementary Table S2). Importantly, both C and C_{iso} exhibit the same surface area (1.27 nm^2), Raman cross-section and affinity to gold surface. Due to the isotope atoms, the SERS peak of C_{iso} at around 941 cm^{-1} was shifted from that of C at around 904 cm^{-1} (Supplementary Figure S8). To demonstrate single-molecule SERS, sub-monolayer of equal moles of C and C_{iso} molecules were adsorbed on the AuNU (C_{iso} C-AuNU) with a surface coverage around 1 molecule per 4 nm^2 . As shown in Figure 4b, one SERS peak appearing at the C range ($900 - 930 \text{ cm}^{-1}$) represented detection of single C molecule, and the emergence of one SERS peak at the C_{iso} range ($930 - 960 \text{ cm}^{-1}$) represented detection event of single C_{iso} molecule. If both peak appeared, it belonged to multi-molecule events.

The BiASERS analysis required >1000 events (SERS spectra) to statistically prove single-molecule SERS (see Methods for data processing details), which actually depended on a sensitive and localized hot spot on the AuNU tip. If the hot spot has a small size to cover only 1 molecule, the probabilities of detecting either one of the molecules are high. If the hot spot became large enough to cover >1 molecules, the probability of detecting both of them increases as well. When the former are larger than the latter in our case of around 1030 events, a BiASERS histogram with more single-molecule events than multi-molecule events (Figure 4c) was established to confirm single-molecule detection.

Figure 4. BiASERS analysis of single-molecule SERS of individual nucleobases. (a) SERS time series of submonolayer $C_{iso}C$ -AuNUs trapped in gold nanoholes. (b) Representative single-molecule spectra and many-molecule spectrum of $C_{iso}C$ -AuNU, respectively. (c) BiASERS distribution of C_{iso} and C out of about 1030 spectra. (d) SERS time series of the submonolayer AG-AuNUs trapped in a nanohole. (e) Typical SERS spectra of single-molecule events and many-molecule events of AG-AuNU. (f) BiASERS distribution of A and G out of about 1200 spectra. (g) SERS time series of the CT-AuNUs trapped in a nanohole. (h) Typical SERS spectra of single-molecule events and many-molecule events of CT-AuNU. (i) BiASERS distribution of C and T from about 1500 collected spectra. All color bars represent the signal-to-baseline intensity of the SERS peaks. The dotted lines indicate SERS peak positions of multilayer nucleobases (black for A or G, red for C or T) adsorbed on silver nanoparticles without bias, respectively, from Ref.⁷

References

1. Koo, K. M.; Sina, A. A. I.; Carrascosa, L. G.; Shiddiky, M. J. A.; Trau, M., DNA-bare gold affinity interactions: mechanism and applications in biosensing. *Analytical Methods* **2015**, *7* (17), 7042-7054.
2. Le Ru, E. C.; Meyer, M.; Etchegoin, P. G., Proof of single-molecule sensitivity in surface enhanced Raman scattering (SERS) by means of a two-analyte technique. *Journal of Physical Chemistry B* **2006**, *110* (4), 1944-1948.
3. Etchegoin, P. G.; Meyer, M.; Blackie, E.; Le Ru, E. C., Statistics of single-molecule surface enhanced Raman scattering signals: Fluctuation analysis with multiple analyte techniques. *Analytical Chemistry* **2007**, *79* (21), 8411-8415.
4. Zrimsek, A. B.; Wong, N. L.; Van Duyne, R. P., Single Molecule Surface-Enhanced Raman Spectroscopy: A Critical Analysis of the Biantalyte versus Isotopologue Proof. *Journal of Physical Chemistry C* **2016**, *120* (9), 5133-5142.
5. Dieringer, J. A.; Lettan, R. B., II; Scheidt, K. A.; Van Duyne, R. P., A frequency domain existence proof of single-molecule surface-enhanced Raman Spectroscopy. *Journal of the American Chemical Society* **2007**, *129* (51), 16249-16256.
6. Kleinman, S. L.; Ringe, E.; Valley, N.; Wustholz, K. L.; Phillips, E.; Scheidt, K. A.; Schatz, G. C.; Van Duyne, R. P., Single-Molecule Surface-Enhanced Raman Spectroscopy of Crystal Violet Isotopologues: Theory and Experiment. *Journal of the American Chemical Society* **2011**, *133* (11), 4115-4122.
7. Madzharova, F.; Heiner, Z.; Guehlke, M.; Kneipp, J., Surface-Enhanced Hyper-Raman Spectra of Adenine, Guanine, Cytosine, Thymine, and Uracil. *Journal of Physical Chemistry C* **2016**, *120* (28), 15415-15423.
8. Zhang, R.; Zhang, Y.; Dong, Z. C.; Jiang, S.; Zhang, C.; Chen, L. G.; Zhang, L.; Liao, Y.; Aizpurua, J.; Luo, Y.; Yang, J. L.; Hou, J. G., Chemical mapping of a single molecule by plasmon-enhanced Raman scattering. *Nature* **2013**, *498* (7452), 82-86.
9. Benz, F.; Schmidt, M. K.; Dreismann, A.; Chikkaraddy, R.; Zhang, Y.; Demetriadou, A.; Carnegie, C.; Ohadi, H.; de Nijs, B.; Esteban, R.; Aizpurua, J.; Baumberg, J. J., Single-molecule optomechanics in "picocavities". *Science* **2016**, *354* (6313), 726-729.
10. Park, K.-D.; Muller, E. A.; Kravtsov, V.; Sass, P. M.; Dreyer, J.; Atkin, J. M.; Raschke, M. B., Variable-Temperature Tip-Enhanced Raman Spectroscopy of Single-Molecule Fluctuations and Dynamics. *Nano Letters* **2016**, *16* (1), 479-487.
11. Titus, E. J.; Weber, M. L.; Stranahan, S. M.; Willets, K. A., Super-Resolution SERS Imaging beyond the Single-Molecule Limit: An Isotope-Edited Approach. *Nano Letters* **2012**, *12* (10), 5103-5110.
12. Willets, K. A.; Wilson, A. J.; Sundaresan, V.; Joshi, P. B., Super-Resolution Imaging and Plasmonics. *Chemical Reviews* **2017**, *117* (11), 7538-7582.
13. Kerman, S.; Chen, C.; Li, Y.; Van Roy, W.; Lagae, L.; Van Dorpe, P., Raman fingerprinting of single dielectric nanoparticles in plasmonic nanopores. *Nanoscale* **2015**, *7* (44), 18612-18618.
14. Sass, J. K.; Neff, H.; Moskovits, M.; Holloway, S., Electric field gradient effects on the spectroscopy of adsorbed molecules. *The Journal of Physical Chemistry* **1981**, *85* (6), 621-623.
15. Chulhai, D. V.; Jensen, L., Determining Molecular Orientation With Surface-Enhanced Raman Scattering Using Inhomogeneous Electric Fields. *Journal of Physical Chemistry C* **2013**, *117* (38), 19622-19631.
16. Moskovits, M.; Dilella, D. P.; Maynard, K. J., SURFACE RAMAN-SPECTROSCOPY OF A NUMBER OF CYCLIC AROMATIC-MOLECULES ADSORBED ON SILVER - SELECTION-RULES AND MOLECULAR-REORIENTATION. *Langmuir* **1988**, *4* (1), 67-76.
17. Belkin, M.; Chao, S.-H.; Jonsson, M. P.; Dekker, C.; Aksimentiev, A., Plasmonic Nanopores for Trapping, Controlling Displacement, and Sequencing of DNA. *Acs Nano* **2015**, *9* (11), 10598-10611.
18. Yang, J.-M.; Pan, Z.-Q.; Qin, F.-F.; Chen, M.; Wang, K.; Xia, X.-H., An in situ SERS study of ionic transport and the Joule heating effect in plasmonic nanopores. *Chemical Communications* **2018**, *54* (94), 13236-13239.
19. Klingsporn, J. M.; Jiang, N.; Pozzi, E. A.; Sonntag, M. D.; Chulhai, D.; Seideman, T.; Jensen, L.; Hersam, M. C.; Van Duyne, R. P., Intramolecular Insight into Adsorbate-Substrate Interactions via Low-Temperature, Ultrahigh-Vacuum Tip-Enhanced Raman Spectroscopy. *Journal of the American Chemical Society* **2014**, *136* (10), 3881-3887.

20. Ichimura, T.; Fujii, S.; Verma, P.; Yano, T.; Inouye, Y.; Kawata, S., Subnanometric Near-Field Raman Investigation in the Vicinity of a Metallic Nanostructure. *Physical Review Letters* **2009**, *102* (18).
21. Watanabe, H.; Ishida, Y.; Hayazawa, N.; Inouye, Y.; Kawata, S., Tip-enhanced near-field Raman analysis of tip-pressurized adenine molecule. *Physical Review B* **2004**, *69* (15).
22. Johnson, C. R.; Asher, S. A., UV RESONANCE RAMAN EXCITATION PROFILES OF L-CYSTINE. *Journal of Raman Spectroscopy* **1987**, *18* (5), 345-349.
23. Blackie, E. J.; Le Ru, E. C.; Etchegoin, P. G., Single-Molecule Surface-Enhanced Raman Spectroscopy of Nonresonant Molecules. *Journal of the American Chemical Society* **2009**, *131* (40), 14466-14472.
24. Clement, J.-E.; Leray, A.; Bouhelier, A.; Finot, E., Spectral pointillism of enhanced Raman scattering for accessing structural and conformational information on single protein. *Physical Chemistry Chemical Physics* **2017**, *19* (1), 458-466.
25. Matteini, P.; Cottat, M.; Tavanti, F.; Panfilova, E.; Scuderi, M.; Nicotra, G.; Menziani, M. C.; Khlebtsov, N.; de Angelis, M.; Pini, R., Site-Selective Surface-Enhanced Raman Detection of Proteins. *ACS Nano* **2017**, *11* (1), 918-926.
26. Xu, L. J.; Zong, C.; Zheng, X. S.; Hu, P.; Feng, J. M.; Ren, B., Label-Free Detection of Native Proteins by Surface-Enhanced Raman Spectroscopy Using Iodide-Modified Nanoparticles. *Analytical Chemistry* **2014**, *86* (4), 2238-2245.
27. Hrelescu, C.; Sau, T. K.; Rogach, A. L.; Jackel, F.; Feldmann, J., Single gold nanostars enhance Raman scattering. *Applied Physics Letters* **2009**, *94* (15).
28. Houry, C. G.; Vo-Dinh, T., Gold Nanostars For Surface-Enhanced Raman Scattering: Synthesis, Characterization and Optimization. *Journal of Physical Chemistry C* **2008**, *112* (48), 18849-18859.
29. Niu, W. X.; Chua, Y. A. A.; Zhang, W. Q.; Huang, H. J.; Lu, X. M., Highly Symmetric Gold Nanostars: Crystallographic Control and Surface-Enhanced Raman Scattering Property. *Journal of the American Chemical Society* **2015**, *137* (33), 10460-10463.
30. Iakhnenko, M.; Feyer, V.; Tsud, N.; Plekan, O.; Wang, F.; Ahmed, M.; Slobodyanyuk, O. V.; Acres, R. G.; Matolin, V.; Prince, K. C., Adsorption of Cytosine and AZA Derivatives of Cytidine on Au Single Crystal Surfaces. *Journal of Physical Chemistry C* **2013**, *117* (36), 18423-18433.
31. Huang, J. A.; Zhao, Y. Q.; Zhang, X. J.; He, L. F.; Wong, T. L.; Chui, Y. S.; Zhang, W. J.; Lee, S. T., Ordered Ag/Si Nanowires Array: Wide-Range Surface-Enhanced Raman Spectroscopy for Reproducible Biomolecule Detection. *Nano Letters* **2013**, *13* (11), 5039-5045.
32. Kim, J. Y.; Han, D.; Crouch, G. M.; Kwon, S. R.; Bohn, P. W., Capture of Single Silver Nanoparticles in Nanopore Arrays Detected by Simultaneous Amperometry and Surface-Enhanced Raman Scattering. *Analytical Chemistry* **2019**, *91* (7), 4568-4576.
33. Otto, C.; Vandenweel, T. J. J.; Demul, F. F. M.; Greve, J., SURFACE-ENHANCED RAMAN-SPECTROSCOPY OF DNA BASES. *Journal of Raman Spectroscopy* **1986**, *17* (3), 289-298.
34. Liu, J., Adsorption of DNA onto gold nanoparticles and graphene oxide: surface science and applications. *Physical Chemistry Chemical Physics* **2012**, *14* (30), 10485-10496.
35. Siiman, O.; Bumm, L. A.; Callaghan, R.; Blatchford, C. G.; Kerker, M., SURFACE-ENHANCED RAMAN-SCATTERING BY CITRATE ON COLLOIDAL SILVER. *Journal of Physical Chemistry* **1983**, *87* (6), 1014-1023.
36. Le Ru, E. C.; Etchegoin, P. G., Single-Molecule Surface-Enhanced Raman Spectroscopy. *Annual Review of Physical Chemistry, Vol 63* **2012**, *63*, 65-87.

REVIEWERS' COMMENTS:

Reviewer #1 (Remarks to the Author):

The authors have taken the advice of the different reviewers very seriously and generated a lot of new data. Not all the data can be explained completely, but that is often also very hard in surface enhanced Raman spectroscopy, especially around the single molecule limit. The additional BiaSERS measurements indeed give strong indications for single molecule SERS observations. That part of the story is made more sound for sure. The relevance of the work for actual applications is still not completely clear to me though, as the technique will mainly allow to detect rather small molecules.

Reviewer #3 (Remarks to the Author):

The authors have satisfactorily addressed my previous comments. The manuscript can be published in the present form.

Reply to referees

Thank you for your comments that help to improve our manuscript. The details of our replies are reported in blue following Reviewers' comments as below:

REVIEWERS' COMMENTS:

Reviewer #1 (Remarks to the Author):

The authors have taken the advice of the different reviewers very seriously and generated a lot of new data. Not all the data can be explained completely, but that is often also very hard in surface enhanced Raman spectroscopy, especially around the single molecule limit. The additional BiaSERS measurements indeed give strong indications for single molecule SERS observations. That part of the story is made more sound for sure. The relevance of the work for actual applications is still not completely clear to me though, as the technique will mainly allow to detect rather small molecules.

Reply: we thank the Reviewer for the encouraging comments. Indeed, a mechanism is needed to unfold large biological molecules such as double-strand DNA so they can adsorb on a gold nanoparticles in a stretched manner for the single-base discrimination by our electro-plasmonic system. While such unfolding mechanism is already available in biological nanopores, its application to solid-state nanopores is being developed. Therefore, we confine our claim of single-base discrimination to oligonucleotides in the manuscript and remain optimistic about analyzing large molecules by integrating the unfolding mechanism with our method.

Reviewer #3 (Remarks to the Author):

The authors have satisfactorily addressed my previous comments. The manuscript can be published in the present form.

Reply: we thank the Reviewer for the recommendation.